# Late time physics of holographic quantum chaos

## Alexander Altland[1*] and Julian Sonner[2†]

**1** Institut für theoretische Physik, Zülpicher Str. 77, 50937 Köln, Germany
**2** Department of Theoretical Physics, University of Geneva,
24 quai Ernest-Ansermet, 1211 Genève 4, Suisse

* alexal@thp.uni-koeln.de, † julian.sonner@unige.ch

## Abstract

Quantum chaotic systems are often defined via the assertion that their spectral statistics coincides with, or is well approximated by, random matrix theory. In this paper we explain how the universal content of random matrix theory emerges as the consequence of a simple symmetry-breaking principle and its associated Goldstone modes. This allows us to write down an effective-field theory (EFT) description of quantum chaotic systems, which is able to control the level statistics up to an accuracy $\mathcal{O}\left(e^{-S}\right)$ with $S$ the entropy. We explain how the EFT description emerges from explicit ensembles, using the example of a matrix model with arbitrary invariant potential, but also when and how it applies to individual quantum systems, without reference to an ensemble. Within AdS/CFT this gives a general framework to express correlations between "different universes" and we explicitly demonstrate the bulk realization of the EFT in minimal string theory where the Goldstone modes are bound states of strings stretching between bulk spectral branes. We discuss the construction of the EFT of quantum chaos also in higher dimensional field theories, as applicable for example for higher-dimensional AdS/CFT dual pairs.

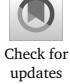
# 1  Introduction and summary

Important progress in lower-dimensional models of AdS/CFT duality has given renewed impetus to contemplate ensemble averages of boundary theories, which either entirely capture the bulk quantum gravity path integral, or at least agree with the latter in an averaged sense.

Ensemble averages are also known to play a crucial role in the understanding of chaotic quantum systems. Generic chaotic quantum systems are believed to exhibit behavior at late times that is well captured by that of a suitable random-matrix ensemble. Suitable here means that there is a small number of universality classes distinguished by the presence or absence of anti-unitary symmetries in the original quantum system and 'late times' means 'later than the Thouless time', whose precise definition we give below. Equivalently, this RMT universality finds its expression in the distribution of energy levels, in the sense that the energy levels of a chaotic quantum system exhibit the same statistical properties as those of the corresponding ensemble.

A key question in both contexts is the relationship between the *individual, fixed* quantum system and the ensemble average that is supposed to capture its chaotic properties. In this work we pursue a set of parallel goals in order to investigate this question, with an eye on its holographic ramifications. Firstly we describe how the universal content of chaotic quantum systems is encapsulated in a simple symmetry-breaking principle, the *breaking of causal symmetry*. Secondly we show that this symmetry breaking principle allows one to select a universal set of light modes – precisely the associated (pseudo–)Goldstone modes – which quantitatively express the RMT physics, both for ensembles as well as individual chaotic quantum systems, in terms of a simple effective (field) theory, well-known in the chaos community

as the Efetov-Wegner sigma-model [1,2]. Thirdly, we argue that in the bulk causal symmetry is broken by a condensate of strings, moving in the presence of D-branes representing the spectral determinants of the boundary theory. As an illustration we give an explicit bulk realization of this scenario in the context of minimal string theory, where the relevant effective theory is equivalent to Kontsevich's cubic matrix model.

The sigma-model approach is tailor-made to control the spectral statistics with a precision that is exponential in the size of the Hilbert space, i.e. sensitive to energy differences of the order

$$\Delta E \sim \mathcal{O}\left(e^{-S}\right). \tag{1.1}$$

These contributions are sometimes referred to as being doubly non-perturbative because they are controlled by factors $e^{-\text{const.}\times e^{S}}$, exponential in the size of the Hilbert space, which itself is exponential in the system size (e.g. $\sim e^{-e^{N}}$ for SYK on $N$ sites.) This is achieved by formulating the calculation of microscopically defined correlation functions in terms of an effective theory of the aforementioned pseudo-Goldstone modes. The large parameter, $\dim \mathcal{H}$, stabilizes two saddle points on the Goldstone mode manifold, connected to each other via a discrete Weyl group symmetry. Within this framework, structures such as the 'dip-ramp-plateau' region ubiquitous in spectral form factors are described as the contributions of non-ergodic fluctuations (dip), an expansion in ergodic fluctuations around the first saddle (ramp), and one around the second (plateau), respectively. The latter two structures are the universal content of the effective field theory of the Goldstone sector. The Weyl group symmetry establishes a connection between the expansions yielding ramp and plateau, respectively, and in this way gives access to deeply non-perturbative information (plateau) once the perturbative contents of a theory (dip/ramp) is under control. Finally, in the deep infrared, $\Delta E \lesssim \mathcal{O}(e^{-e^{S}})$, the ergodic mode becomes massless, full integration over its nonlinear target manifold restores the causal symmetry, and in this way resolves the rigidity of level repulsion in the chaotic many body spectrum in quantitative agreement with the results of random matrix theory (RMT).

In cases where a theory is exactly given by a matrix model (say of matrix dimension $L \times L$), for example in 2D gravity [3], or the recent reformulation of JT gravity in terms of matrix integrals [4], the above approach improves on previous treatments in that it gives analytical control over non-perturbative physics, i.e. contribution of $\mathcal{O}\left(e^{-L}\right)$. In situations where the theory is not exactly equivalent to a matrix model, for example in the canonical holographic example of the $\mathcal{N} = 4$ theory in $3 + 1$ dimensions, the sigma-model approach advocated in our work gives a framework to understand in what sense random-matrix physics is a good approximation, and to control spectral correlations non-perturbatively in terms of an effective field theory.

The general idea of causal symmetry itself is quite simple: in order to extract information about spectral correlations, it is necessary to consider correlation functions between Green functions of opposite causality, that is to say between advanced and retarded propagators. It is then convenient to develop a (path-)integral representation of these correlation functions by introducing auxiliary degrees of freedoms, at first separately for the advanced as well as the retarded sector. However, the resulting system turns out to be invariant under continuous general linear transformations which rotate advanced into retarded degrees of freedom (and vice versa), the causal symmetry. Furthermore, in applications of interest to us, this symmetry is broken both explicitly by a small amount and spontaneously by the saddle-point solution(s).

In such cases, one may describe the long time or low energy physics of the system in terms of an effective field theory assuming the symbolic form,

$$\mathcal{Z} = \int DQ \, e^{-S[Q]}. \tag{1.2}$$

Here, the integration is over fields, seen as maps from a base space $M$ to the target space $T$, $Q : M \rightarrow T, x \mapsto Q(x)$, where $T$ is a low-dimensional nonlinear target manifold universally

determined by the symmetry breaking. For example, in many cases of interest $Q(x)$ is simply a $4 \times 4$ graded matrix, and $T$ the graded coset space $U(2|2)/U(1|1) \times U(1|1)$ (hence the classification as nonlinear sigma model.) By contrast, the realization of the base space depends on the application. For example, in the case of matrix models, the integral is over matrices $Q$, with no position dependence. In the classic applications of the theory to disordered electronic systems, $Q(x)$ becomes a space-dependent matrix field, and $S[Q]$ an action containing spatial gradient terms $\int \mathrm{STr}(\nabla Q(x))^2$ (describing diffusive dynamics) in addition to the explicit symmetry breaking mass term $\int \mathrm{STr}(\omega Q)$. However, more relevant to holographic applications are many body realizations of chaos, where $M$ reflects the full structure of the underlying many-body Hilbert space. Given this potentially highly complicated structure it is fortunate that typically (that is in theories that do not exhibit many-body localization) there exists a threshold energy, the so-called Thouless energy, below which fluctuations inhomogeneous on $M$ are energetically disfavoured, so that the field integral collapses to one over a homogeneous 'mean field' configuration $Q(x) = Q$. We thus conclude that, for these energies, the action reduces to that of the random matrix models, thereby demonstrating RMT universality at low energies from the perspective of the EFT. The appearance of the scale $L$ in the exponent, which is a robust construction principle of the present approach, means that the theory is 'semiclassical' for energies $\Delta E \ll L^{-1}$, and capable of resolving 'doubly non-perturbative' structures for $\Delta E \sim L^{-1}$. We remark that, as will become clear below, the notion of 'low-energy' in the EFT we develop here, refers to small differences of energies in the spectrum of the original system.

The essence of the sigma model approach is a reduction of theories defined in $e^S$-dimensional Hilbert spaces to effective theories on much lower dimensional 'flavor' manifolds. Both the realization of the flavor manifold and its exact dimensionality (which however remains always of $\mathcal{O}(1)$ in terms of $L$) depend on the symmetries of the parent theory, and on the details of the correlation functions at hand, in a well-understood way.

As we will reason later in the text, this reduction affords a beautifully simple bulk interpretation: the flavor manifold can be identified with the effective degrees of freedom of a sector of open strings which stretch between a small number of 'flavor branes', while the pseudo-Goldstone modes are mesonic strings which begin and end on one of these flavor branes. Again the nature and the number of 'flavor branes' depends on the observables one wants to compute, and is in exact correspondence with the aforementioned flavor structure of the boundary theory.

In the rest of the paper, we will discuss the contents prosaically summarized above in more mathematical language. Much of the paper addresses contents well familiar in either the string theory or the condensed matter/chaos community, but hardly any of it in both. We have tried to make the text accessible to readers from both camps, which explains a degree of redundancy and the presence of material which may be skipped by experts. In section 2 we introduce standard diagnostic observables probing the physics of chaotic systems, their representation in terms of graded field integrals, the principles of the above symmetry breaking mechanism, and that of the field theory defined by it. While the discussion of this section applies to a wide range of settings, section 3 takes a complementary perspective and considers matrix theories — with invariant yet general (not necessarily Gaussian) distribution — as a concrete case study. This section intentionally goes into some technical detail. It can, but need not be read before the more exploratory section 4, where we address causal symmetry breaking from the perspective of the bulk. We present a discussion of some open issues Section 5, followed by Appendices going over salient details of impurity diagrams as well as the relation of the superymmetry technique used in the bulk of this paper and the replica approach.

## 2 Spectral probes

### 2.1 Time scales

The prime focus of this work rests on late time chaotic properties, so it will be useful to put the relevant scales in context. In order to do so, let us define a few important quantities, starting with the density of energy levels

$$\rho(E) = \frac{1}{L} \sum_{i=1}^{L} \delta(E - E_i), \tag{2.1}$$

where $L = \dim\mathcal{H}$ is the dimension of Hilbert space. Since the quantum system of interest will have a very large number of such levels, it makes sense to speak of the typical separation at energy $E$, the mean level spacing

$$\Delta(E) := \langle \rho(E) \rangle^{-1}, \tag{2.2}$$

where the average denoted by angular brackets may be over energy windows of a given Hamiltonian, a set of parameters, or as will often be the case, over an ensemble of Hamiltonians. The mean level spacing defines the smallest energy scale in the problem, and its inverse, the Heisenberg time, $t_H = 2\pi\Delta^{-1}$ the largest time scale. (Referring to the emergence of a 'plateau' in the form factor Eq. (2.7) of Gaussian unitary random matrices at this scale, $t_H$ has been dubbed the 'plateau time'. However, given that in other symmetry classes there are no such signatures in the form factor, this may be a bit of a misnomer.) Following standard conventions in the field, we frequently measure energy differences $\omega = \mathcal{O}(\Delta)$ and their conjugate time scales $t$ in dimensionless variables

$$s \equiv \frac{\pi\omega}{\Delta}, \qquad \tau \equiv \frac{t}{t_H}. \tag{2.3}$$

We next outline some canonical time scales encountered in quantum chaotic systems, in order to better define the regime of interest of this work, namely times of parametric order $t_H$.

Firstly, let us generically assume that there is a parameter, such as $\hbar$ or $N$, so that the system is semi-classical for $\hbar, N^{-1} \to 0$. A typical chaotic quantum system then has characteristic imprints resulting from the following scales:

**Early time chaos**

1. *Lyapunov time, $t_L = \lambda^{-1}$*: Many chaotic systems are defined with reference to a semiclassical parameter $\hbar$ such that they become classical in the limit $\hbar \to 0$. In these cases, the time scale $t_L$ characterizes the exponential divergence $\delta x \sim e^{\lambda t}$ of initially close phase-space trajectories. In systems without straightforward classical limit, such as spin 1/2 chains, time scales playing an analogous role may be defined in terms of observables showing early time exponential instabilities.

2. *Ehrenfest time $t_E \sim \lambda^{-1} \log 1/\hbar$*: this quantum time scale is attained when a minimum-uncertainty wavepacket localised to within a Planck cell in phase space has spread to a width of order one. The prefactor of $\lambda^{-1}$ is due to the semiclassical propagation of the wavepacket (i.e. retains an imprint of classical Lyapunov chaos), while the factor $\hbar$ inside the log comes from the quantum spread. This timescale is also known, especially in the black-hole context, as the *scrambling time $t_s$*. In cases without orthodox semiclassical parameter, $\hbar$, the scrambling time is reached when observables probing early time instabilities have become 'large', for instance at $t_E \sim \lambda^{-1} \ln(N)$.

3. *Ergodic time (aka Thouless time) $t_E$*: the time scale beyond which a chaotic flow uniformly covers the classical energy shell in phase space. For example, in a diffusive metal of linear extent $\lambda$ and diffusion constant $D$, this would be the time $t_E \approx D/\lambda^2$ it takes to diffusively explore the system. More generally, $t_E$ sets the scale beyond which RMT behavior is attained. We caution that a Thouless time need not necessarily exist (such as in infinitely extended systems, or systems which do not quantum-thermalize in the long time limit), or can be be a tricky affair (such as in the SYK model, where the fidelity to RMT depends on the observable under consideration).

**Late time chaos**

4. *Times exceeding the ergodic time $t > t_E$:* The physics described in this regime is what is often referred to as the 'ramp' phase, associated with spectral rigidity of chaotic quantum systems. However, for $\tau \sim \mathcal{O}(1)$, necessarily non-perturbative physics sets in. This happens at the

5. *Heisenberg time $t_H = 2\pi\Delta^{-1}$*: the behavior associated with this time scale is what is known as the 'plateau' phase when speaking of certain observables, such as two-point functions or the spectral form factor. The physical behavior in this regime is non-perturbative in $\Delta$, but straightforward to capture within the late-time effective description expanded upon in this work.

To orient the reader, let us indicate these timescales on the example of the Majorana SYK model [5,6] on $N$ sites, which has Hilbert space dimension $L = 2^{N/2}$, dual to black holes in two dimensional anti-de Sitter space [5,7]. A seminal computation by Kitaev demonstrated that the scrambling time for this model takes the form $t_s = \frac{\beta}{2\pi}\log(N)$, indicating that $N^{-1}$ assumes the role of the semiclassical parameter. The mean level spacing $\Delta \sim e^{-N}$ is exponentially small in $N$, the Heisenberg time scales like $t_H \sim e^N$, and the 'doubly non-perturbative' effects appearing when $\tau = 1$ are of order $e^{-e^N}$. These have been the object of much recent interest [4], and as part of this work we review how they are explicitly and analytically computable within the EFT approach [8]. This is in contrast to the picture in [4], where the existence of such effects is inferred from the asymptotic behavior of a perturbative expansion in $\Delta$, while the non-perturbative completion itself remains inaccessible.

## 2.2 Resolvents and determinants

Let us now introduce a set of observables well suited to characterise spectral properties of a quantum system of interest. We start by defining the trace of the resolvent

$$W(z^{\pm}) = \left\langle \text{Tr}\frac{1}{z^{\pm} - H} \right\rangle = \left\langle \text{Tr}G(z^{\pm}) \right\rangle, \tag{2.4}$$

where for the time being the brackets indicate an average either over an energy window or an ensemble of Hamiltonians and we have introduced the notation $z^{\pm}$ indicating the addition of a small positive or negative imaginary part the energy argument. The utility of the spectral resolvent is in its simple relationship with the spectral density, namely $\rho(z) = \mp\frac{1}{\pi}\text{Im}W(z \pm i\delta)$. Mostly we will be interested in higher spectral correlation functions, and correspondingly in objects of the form

$$W(z_1, z_2, \ldots z_k) = \left\langle \text{Tr}G(z_1)\text{Tr}G(z_2)\cdots\text{Tr}G(z_n) \right\rangle, \tag{2.5}$$

where each energy argument could have either a small positive or negative imaginary part, and which shall often be indicated as $z_i^{\pm}$. A particularly interesting such quantity is the spectral

form factor, which is defined via the spectral correlation function

$$R_2(\omega) = \Delta^2 \left\langle \rho(E + \frac{\omega}{2})\rho(E - \frac{\omega}{2}) \right\rangle_c , \tag{2.6}$$

(we denote the connected part ($\langle AB \rangle_c = \langle AB \rangle - \langle A \rangle \langle B \rangle$) of any observable with subscript 'c'), such that

$$K(\tau) = \frac{1}{\Delta} \int d\omega R_2(\omega) e^{-i\frac{2\pi\omega}{\Delta}\tau} . \tag{2.7}$$

Note that the exponent is written in a natural way in terms of times normalized by the inverse level spacing. Having discussed these characteristic probes, we now introduce a class of auxiliary quantities, which will turn out to be the most convenient objects on which we base our framework. These take the form of ratios of determinants and we are principally interested in two cases, namely

$$\mathcal{Z}_{(2)}(\hat{z}) = \frac{\det(z_1 - H)}{\det(z_2 - H)}, \qquad \mathcal{Z}_{(4)}(\hat{z}) = \frac{\det(z_1 - H)\det(z_2 - H)}{\det(z_3 - H)\det(z_4 - H)} . \tag{2.8}$$

By a slight abuse of notation, in each case the matrix $\hat{z}$ is taken to be the diagonal matrix of ordered energy arguments, e.g. $\hat{z} = \mathrm{diag}(z_1, z_2)$ in the first case. The dimension of this matrix will always be clear from the context, and the utility of arranging energies in matrix form will emerge in our subsequent developments. Let us briefly pause and note that careful attention needs to be placed on the infinitesimal imaginary parts given to the energy arguments, in the *denominators*, where they determine the pole structure of the spectral determinants. Specifically, the spectral determinant $\mathcal{Z}_{(2)}$ allows us to extract the spectral density via the resolvent, viz.

$$W(z) = \partial_{z_2} \mathcal{Z}_{(2)}(\hat{z}) \Big|_{z_1 = z_2 = z} \quad \Leftrightarrow \quad \rho(z) = \mp \frac{1}{\pi L} \mathrm{Im}\, \partial_{z_2} \left\langle \mathcal{Z}_{(2)}(\hat{z}) \right\rangle \Big|_{z_1 = z_2 = E \pm i0} . \tag{2.9}$$

On the other hand, the spectral two point function requires the use of the ratio $\mathcal{Z}_{(4)}$:

$$\langle \rho(E)\rho(E') \rangle = \frac{\mathrm{Re}}{\pi^2 L^2} \frac{\partial^2}{\partial z_3 \partial z_4} \left\langle \mathcal{Z}_{(4)}(z_1, z_2, z_3^+, z_4^-) \right\rangle \Big|_{\substack{z_1 = z_3^+ = E \\ z_2 = z_4^- = E'}} . \tag{2.10}$$

The spectral determinant $\mathcal{Z}_{(4)}$ possesses an interesting *Weyl symmetry* under the exchange of $z_1 \leftrightarrow z_2$, which leaves the spectral correlation function unchanged. As we will discuss later, this discrete symmetry is key to the understanding of the non-perturbative structure of the spectrum. It is called 'Weyl symmetry' because in the later representation of the problem as a coset matrix integral the reflection $z_1 \leftrightarrow z_2$ translates to a Weyl group symmetry in the mathematical sense on the symmetry group of the integral.

## 2.3 Causal symmetry

A very useful way to rewrite the ratios (2.8) is by means of a graded Gaussian integral

$$\mathcal{Z}_{(4)}(\hat{z}) = \int e^{-i\bar{\psi}(\hat{z}-H)\psi} d(\psi, \bar{\psi}), \tag{2.11}$$

where $\psi \in \mathcal{V}$ is a $4L$ dimensional graded vector with $2L$ Grassmannian components and $2L$ ordinary c-number components. Here, the Grassman integrals produce the determinant factors in the numerator, while the c-number integrals produce those in the denominator. It should

also be clear how to write the corresponding expression of a ratio $\mathcal{Z}_{(2n)}$ in terms of a $2nL$-dimensional graded integral, although we shall not be needing the higher $n > 2$ cases in the present work. The detailed structure of the graded vector space is associated to different physical aspects of our problem and takes the form $\mathcal{V} = \mathcal{H} \otimes \mathcal{V}_F$, where $\mathcal{H}$ is the Hilbert space our Hamiltonian $H$ acts on, and we refer to $\mathcal{V}_F$ as "flavor space". We note that in this context the Hilbert space itself is often referred to as "color space". We will often use this terminology in this work, but caution the reader not to confuse this usage of "color" with the common usage of color symmetry in Yang-Mills theory. On the other hand, the flavor-structure is dictated by the requirement that each determinant must come with an inverse determinant so as to ensure normalization of the final physical quantities. This introduces a $\mathbb{Z}_2$ grading to flavor space[1]. Finally, in the cases of interest we have advanced and retarded sectors which each add one more factor to the tensor product making up flavor space.

Let us now describe more closely the structure of the integral (2.11) above: as mentioned, each of the fields $\psi = \{\psi_\mu^a\}$ carries a Hilbert-space index $\mu$ as well as a flavor index $a$, suppressed above for notational transparency. The action of the Gaussian integral is subject to a GL($2L|2L$) symmetry in the fundamental representation,

$$\text{GL}(2L|2L) \qquad \Rightarrow \qquad \psi \to g\psi, \quad \bar{\psi} \to \bar{\psi}g^{-1}, \tag{2.12}$$

weakly broken by the energy matrix $\hat{z}$, and strongly broken by the Hamiltonian $H$. (We here use standard $(\cdot|\cdot)$ notation in referring to group representations in graded spaces.) The origins of this symmetry and its (spontaneous) breaking in the causal sector will be the main guiding principle in the construction of this paper. Secondly, the index $a$ plays a double role. It labels both fermionic and bosonic components, as well as advanced and retarded components, distinguished by their respective imaginary offsets. One may thus also write $a = (\sigma, s)$, where $s = \pm$ labels the component in the advanced-retarded basis and $\sigma$ denotes the fermion-boson grading. Thirdly, the action of conjugation (see section 3, after Eq. (3.1) for the detailed definition) $\bar{\psi}$ together with the infinitesimal imaginary offsets $\delta_a$ ensure convergence in the bosonic sector.

The field theory approach to quantum chaos takes the exact representation (2.11) as the starting point towards the construction of an effective low energy theory describing the system at large time scales. Here 'low energy' refers energy differences of order $|z_i - z_j| \sim \Delta$, assumed to be of the order of the quantum energy level spacing (1.1). In certain examples, these theories can be derived from first principles where both the final form of the effective action, and the details of the construction are specific to the physical system at hand. However, we here reason from the vantage point of symmetries, which lets the underlying structures stand out, and defines generally applicable construction principles. Specifically, symmetries determine the target spaces of the effective field theories, the essential strategy of their derivation, their operator contents, and the universal physics of the ergodic phase (where one exists.) In the following, we introduce these symmetries and their manifestations in the effective low energy theory from a birds eye perspective. This discussion is complemented in section 3 by an exemplary derivation of the effective field theory for the simple case where $H$ is drawn from a matrix ensemble. In section 4 we give an explicit bulk realization, again for what is arguably one of the simplest examples, namely $(2, q)$ minimal string theory, and draw broader conclusions for holographic duality.

---

[1]Let us note that the normalization of final results can also be achieved by means of the replica trick, which would lead to a much bigger, but purely bosonic flavor space. We shall comment on this alternative point of view from time to time.

## 2.4 Ensemble vs. individual system

*Unus pro omnibus, omnes pro uno*
(The unofficial motto of Switzerland)

We have at various points stated that causal symmetry breaking is the universal description of quantum chaotic systems. However, this begs the question to what extent individual chaotic systems display this 'universal' behavior. Our approach to quantum chaos is inherently statistical, describing systems in terms of correlation functions which either directly or effectively make reference to (parametric) ensembles. For an individual system the notion of statistics does not exist in a strict sense[2]. On the other hand we know that if we average over an ensemble of microscopically different but macroscopically identical systems, universal statistics emerges. On this basis, the expectation is that in spectral correlation functions computed for individual chaotic systems, smooth backgrounds exhibiting RMT behavior appear superimposed with high frequency noise — see Ref. [9] for a case study demonstrating this phenomenon. For a system of Hilbert space dimension $L$, the noise amplitude scales with $\sim L^{-1/2}$, masking the rapid universal decay of spectral correlations already for small energy differences. However, these fluctuations are extremely rapid, making them susceptible to dephasing under any mild averaging. This has the effect that signal smoothing by *any* continuous averaging protocol, over external system parameters, 'disorder', or even the value of $\hbar$, efficiently eliminates it and allows the universal content to emerge. These observations comport well with recent work that proposes to define statistical ensembles for AdS$_3$ gravity by averaging over a set of moduli of the compactification [10–13], which serve as the small set of smoothing parameters necessary to bring out the EFT behavior.

If one wants to strictly confront individual chaotic systems, without any mild averaging over a small set of parameters whatsoever, the EFT can be stabilized by an average over energy [14–16]. A subsequent expansion in smooth field fluctuations then yields an action which, by design, disposes with the information on high frequency noise. Its capacity to yield universal information beyond the RMT limit has been demonstrated on a number of case studies, including the field theory approach to quantum graphs [17], or to nonperturbative localization phenomena in the quantum standard map [18]. While slightly going against our EFT logic, it may be instructive to apply such an approach to one of our canon of field theories with holographic duals in higher dimensions, e.g. ABJM theory or the $\mathcal{N} = 4$ SYM theory.

## 2.5 The effective field theory of quantum chaos

Having qualitatively outlined the main ideas going into developing an effective theory of universal spectral correlations in quantum chaotic systems, we now delve into the conceptual steps involved in its construction in some more detail.

1. *Broken symmetry and the analytic structure of the resolvent:* consider the resolvent $G(z) \equiv (z - H)^{-1}$ of a chaotic Hamiltonian, where we will assume that $L = \dim \mathcal{H} \gg 1$. For example, considering the case where $H = H(v)$ depends on randomness (symbolically represented by the variable $v$) and the averaging in Equation (2.4) is over a distribution $P(v)$ of the latter. Prior to averaging, the resolvent has poles at the discrete eigenvalues of $H(v)$, implying that $\operatorname{Im} \operatorname{tr}(G(z)) = 0$ almost everywhere for $z = \epsilon \pm i0$. By contrast, the average resolvent $\langle G(z) \rangle$ has a branch cut inside the spectral support of $H$ along the real axis, where $\operatorname{Im} \operatorname{tr} \langle G(z) \rangle = \pm \mathcal{O}(\rho(z))$, the sign being uniquely determined by that of the infinitesimal imaginary part of the energy argument, $\operatorname{Im}(z)$. This illustrates the simplest

---

[2]unless one admits the distribution of its energy levels along the real axis as a discrete measure — an approach that worked miraculously well since the early days of the field when the resonance spectra of individual heavy nuclei were put in relation to RMT spectral statistics.

instance of a symmetry breaking scenario characterized by an amplification $\pm i0 \to \pm i\rho$ of the infinitesimal imaginary energy increments to a large and finite value, proportional to the averaged spectral density, $\rho(z)$.

2. *Pattern of symmetry breaking:* the consequences of the symmetry breaking in the effective theory become apparent when we discuss it in the context of the transformation group Eq. (2.12). Since $H$ represents an explicit and strong symmetry breaking in the 'color' sector, only transformations in $\mathrm{GL}(2L|2L) \to \mathrm{GL}(2|2) \times \mathbb{1}_L \equiv \mathrm{GL}(2|2)$ can be symmetries of the late time physics.. The above breaking mechanism collapses this flavor symmetry group to $\mathrm{GL}(1|1) \times \mathrm{GL}(1|1)$ — two dimensional transformations between bosonic and fermionic degrees of freedom acting separately in the sector of retarded and advanced indices, $s = \pm$. The transformation between the two causal sectors are thus spontaneously broken, whence the term 'causal symmetry breaking'. Furthermore the symmetry is explicitly, but weakly broken by the differences in the energy arguments entering the diagonal matrix $\hat{z}$. We thus conclude that the degrees of freedom essential to the low energy physics of the system are flavor Goldstone modes drawn from the manifold[3]

$$\mathcal{M} = \mathrm{GL}(2|2)/\left(\mathrm{GL}(1|1) \times \mathrm{GL}(1|1)\right). \tag{2.13}$$

3. *Goldstone modes:* since the physics is effectively dominated by the light modes, the effective theory will be given by an integral over the soft manifold. The convergence of this integral requires a reduction to the coset space $\mathrm{U}(2|2)/\mathrm{U}(1|1) \times \mathrm{U}(1|1)$, where the bosonic (fermionic) sector of $U(2|2)$ is the (pseudo)unitary group in two dimensions. Geometrically, the bosonic sector of the coset is a two-dimensional hyperboloid, and the fermionic one a two-dimensional sphere. A convenient way to represent this manifold is in terms of the matrix field

$$Q = T\tau_3 T^{-1}, \qquad T \in \mathrm{U}(2|2), \tag{2.14}$$

where $\tau_3$ is the Pauli matrix in causal space $(\tau_3)^{ss'} = (-)^s \delta^{ss'}$, and the action of $T$ by conjugation respects the residual $\mathrm{U}(1|1) \times \mathrm{U}(1|1)$ symmetry. More generally, the Goldstone modes $Q(x)$ are fluctuating degrees of freedom, where $x$ parameterizes the base space of the effective theory. In this case, the Goldstone modes are constructed by writing $T$ as a product of an element in the full group $G$ times an element of the preserved subgroup $H$. This is achieved by letting $T(x) \to T(x)H(x)$, $H(x) \in \mathrm{U}(1|1) \times \mathrm{U}(1|1)$ so that $H$ becomes an exact local symmetry, and $T(x) \to T_0 T(x)$, $T_0 \in \mathrm{U}(2|2)$ a global one, weakly broken by the differences of the energy arguments entering the argument $\hat{z}$.

4. *Effective action:* Historically, the present form of nonlinear sigma models systems was pioneered in their application to the physics of disordered metals. There, $x$ is a real space coordinate, and the action assumes the form

$$S[Q] = -i\nu \int \mathrm{STr}(\hat{z}Q)\, dV + F^2 \int \mathrm{STr}\left(\nabla_i Q \nabla^i Q\right) dV + \cdots, \tag{2.15}$$

where 'STr' is the matrix trace generalized to 'supermatrices' containing commuting and anti-commuting elements.[4] The dimensionful quantities $\nu$ and $F^2$ can be identified (see section 3) with the density of states per volume, $\rho(z)$, and the diffusion constant $D$

---

[3]The presence of anti-unitary symmetries in the microscopic theory, such as time reversal, charge conjugation, or chiral symmetries, restricts the set of symmetry-compatible continuous transformations and changes the Goldstone mode manifold. However, for simplicity, we focus on the simplest setting, where $H$ is just hermitian.

[4]For a supermatrix $F = \left(\begin{smallmatrix} a & \sigma \\ \tau & b \end{smallmatrix}\right)$ with bosonic blocks $a, b$ and fermionc blocks $\sigma, \tau$ is defined as $\mathrm{STr}\, F = \mathrm{Tr}\, a - \mathrm{Tr}\, b$.

respectively. Note the structural similarity to the chiral Lagrangian of QCD [19] (albeit with a different symmetry-breaking pattern) which shows that the diffusion term enters in the same way into the effective field theory of chaos as the Pion decay constant enters into the chiral effective Lagrangian[5]. However, unlike the QCD Lagrangian, $S[Q]$ does not describe a dynamical theory, the formal reason being that we consider correlations at fixed energy, so that time is effectively frozen out.

However, most relevant to the present context are recent extensions of the formalism to interacting systems such as the SYK model [8, 20, 21]. In these cases, the coordinates $x \to n$ label the $L \sim e^S$ discrete basis states of the many-body Hilbert space, and the action assumes the symbolic form

$$S[Q] = -i\nu \sum_n \text{STr}(\hat{z}Q) + \sum_{n,m} W_{nm} \text{STr}(Q_n Q_m) + \cdots, \qquad (2.16)$$

where $\nu$ is now the density of states per lattice site, and $W_{nm}$ are the lattice hopping matrix elements defined by the underlying many body Hamiltonian. (For example, the four Majorana interaction defining the SYK Hamiltonian [5] can change up to four fermion occupation numbers, making for a range four hopping operator $W_{nm}$.) Depending on the strength of the hopping elements, the above model describes a thermalizing phase where below a finite Thouless energy the uniform mode $Q_n \to Q$ dominates (this happens, e.g., in the application to the SYK model), or a 'many body localized' phase with strong independent fluctuations of $Q_n$ (see Ref. [22] for review.)

In ergodic regimes, both the action describing single particle systems Eq. (2.15), and many body systems, Eq. (2.16) collapse to the zero mode action

$$S(Q) = -i\frac{\pi}{2\Delta} \text{STr}(Q\hat{z}), \qquad (2.17)$$

where $Q(x) = Q_n = Q$ is the homogeneous zero mode and the level spacing defined as $\Delta^{-1} = \frac{2}{\pi}\nu V = \frac{2}{\pi}\nu \sum_n 1$ defined through the effective dimension of Hilbert space. In this regime, the systems are physically equivalent to random matrix models (which are likewise described by the action (2.17) as we will demonstrate in section 3):

5. *Extracting the random-matrix physics:* The partition sum describing a chaotic quantum system in the ergodic regime assumes the form

$$\langle \mathcal{Z}(\hat{z}) \rangle = \int dQ\, e^{-S(Q)}, \qquad (2.18)$$

where the action is given by Eq. (2.17) and the integral is over a single instance of the matrices Eq. (2.14). Thinking of $Q$ as elements of a generalized sphere, and $dQ$ the corresponding invariant measure (section 3 will provide the details), the action is that of a 'magnetic field' of strength $\sim s = \pi\omega/\Delta$. That action comes with two saddle points, a stable one on the north pole with action 0, and an unstable one at the south pole with action $2is$. The fluctuations around both poles are suppressed in the same parameter $s$. In this way, we can understand how the present formalism produces Eq. (2.26) for the spectral function. In the next section, we add some physical contents to the mathematical structure of Eq. (2.18). We will discuss the semiclassical interpretation around the two saddles, its connection to the ramp-plateau profile of spectral correlations, and the idea of a holographic bulk interpretation of these structures.

---

[5]One can write down higher order terms in the symmetry breaking parameter such as $\text{STr}(Q\hat{z}Q\hat{z})$ by promoting $\hat{z}$ to a spurion field. Higher powers of $\hat{z}$ are then suppressed by powers of $\nu/F^4$ leaving us with having to deal only with the most relevant one we wrote.

## 2.6 The ergodic sector of the EFT and its topological expansion

In most holographic applications to date, one is interested in systems which have an ergodic limit (i.e. not many-body localized) and in this work in particular, we are interested in the universal behavior of the ergodic limit.

### Saddle points and Weyl symmetry

Let us thus take a closer look at the EFT in its ergodic phase, i.e. the integral Eq. (2.18), starting at first in the case $s \gtrsim 1$ of energy splittings exceeding the level spacing. In this case, the integral will be dominated by small fluctuations around its stationary points, $\bar{Q}$ where the latter are identified by stationarity under variations, $\delta S[\bar{Q}] = 0$. Using the representation Eq. (2.14), it is straightforward to verify that the stationarity condition is equivalent to $[\bar{Q}, \hat{z}] = 0$, or matrix-diagonality of $\bar{Q}$. A somewhat closer analysis shows that of the four saddle points compatible with the unit-modularity of the eigenvalues $\mathrm{spec}(Q) = \{1, 1, -1, -1\}$ only two are compatible with the manifold structure.

To understand this in an intuitive way, we recall that the commuting variables contained in $Q$ parameterize $H^2 \times S^2$, the product of a (non-compact) hyperboloid and a (compact) two sphere. We can conveniently parametrize the compact sector $Q^{\mathrm{ff}} = T^{\mathrm{ff}} \tau_3 T^{\mathrm{ff}-1} = n_i \tau_i$ in terms of the unit vector, $\mathbf{n}$. Then the two saddle point are the *standard saddle*, $Q_0 = \mathrm{bdiag}(\tau_3, \tau_3)$, and the *Altshuler-Andreev saddle* [23] $Q_{\mathrm{AA}} \equiv \mathrm{diag}(\tau_3^{\mathrm{bb}}, -\tau_3^{\mathrm{ff}})$. In terms of the previously defined unit vector, we see that the standard saddle is located at the north pole of the sphere $\mathbf{n} = (0, 0, 1)$ and the Altshuler-Andreev saddle is at the south pole, $\mathbf{n} = (0, 0, -1)$. Noting that the spectral probes of interest, (2.8), are computed in the particular configuration $\mathrm{Re}\, z_1 = -\mathrm{Re}\, z_2 = \mathrm{Re}\, z_3 = -\mathrm{Re}\, z_4 = -\omega/2$, we obtain the corresponding actions as $S[Q_0] = 0$ and $S[Q_{\mathrm{AA}}] = -2is$, respectively.

Finally, notice that the map $Q_0 \to T_W Q_0 T_W^{-1} \equiv Q_{\mathrm{AA}}$ permuting diagonal matrix elements is an element of the Weyl group of the underlying supergroup structure. The above operation is equivalently described by a transformation of energy arguments, $T_W^{-1} \hat{z} T_W \equiv \hat{z}'$, which permutes $z_1 \leftrightarrow z_2$, an exchange introduced in connection with (2.10) as a Weyl symmetry of the spectral determinant. Writing the Weyl symmetry transformation as

$$\mathrm{STr}(Q_{\mathrm{AA}} \hat{z}) = \mathrm{STr}(Q_0 \hat{z}'), \tag{2.19}$$

we can see that the Weyl exchange of the energy arguments transforms the stationary configuration from the standard to the AA saddle. This raises the possibility that one can 'bootstrap' the non-perturbative content of the Altshuler-Andreev saddle, by using the Weyl group transformation on the theory written around the standard saddle. Indeed, in connection with periodic orbit theory [24] permutations of energies in the spectral determinant have been applied to access non-perturbative information from perturbative orbit expansions. (See remarks below Eq. (2.26) for the connection of this operation to the Riemann-Siegel lookalike hypothesis [25] for the extension of semiclassical analysis.)

In the following we take a closer look at the contribution of these two stationary points to spectral correlation function sand their interpretation in a holographic bulk language.

### Perturbative contributions: wormholes and baby universes

Before delving into the bulk story in section 4, let us explain qualitatively how the formalism naturally produces correlations of the type associated with Euclidean wormholes. As is well known from the study of low-energy QCD — or indeed any other context where pseudo-Goldstone bosons dominate the physics — the physical manifestation of the coset are the

Goldstone modes (aka pions) associated to the generators of the broken symmetry. Let us schematically write

$$T = \exp(W), \qquad W = \begin{pmatrix} & B \\ \tilde{B} & \end{pmatrix}, \tag{2.20}$$

where $W$ is the matrix of pion fields, expanded in the broken generators $B$ and $\tilde{B}$, which individually are (1|1) supermatrices. To leading order in these generators, the action takes the form $S[B, \tilde{B}] = -2is \, \mathrm{STr}(B\tilde{B})$, where for the time being we expand around the 'standard saddle'. We have written the exponent to leading (quadratic) order only, which is justified by the EFT logic and which we will correct systematically. Expanding around this quadratic limit, we are led to consider matrix integral averages of the type

$$\langle \dots \rangle \equiv \int d(B, \tilde{B}) \, e^{-2is \, \mathrm{STr}(B\tilde{B})} (\dots), \tag{2.21}$$

where the operator insertions are similarly built up from the $B, \tilde{B}$ matrices. Specifically, in the computation of the two-point function at leading order we are instructed to compute the expectation value $\langle \mathrm{STr}(B\tilde{B}P^{\mathrm{f}})\mathrm{STr}(\tilde{B}BP^{\mathrm{f}}) \rangle$, where the matrices $P^{\mathrm{f}} = \{\delta_{\sigma,\mathrm{f}}\delta_{\sigma',\mathrm{f}}\}$ project onto the Grassmann sector. The matrices $B$ and $\tilde{B}$ each have one index acting in the retarded sector and one index acting in the advanced sector, and the order of multiplication in the preceding expectation value effectively projects once on the advanced (red) and once on the retarded sector (blue), as required by the prescription in (2.8). We may then represent the matrices $B, \tilde{B}$ using double line notation, the Wick contraction of these matrices with the Gaussian weight Eq. (2.21) can be represented as

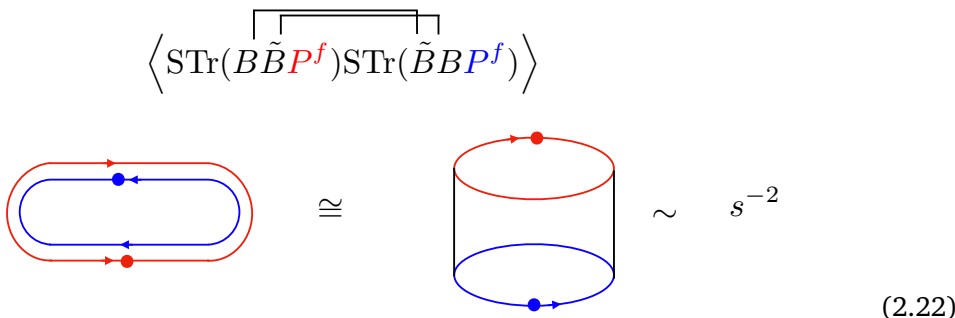

$$\tag{2.22}$$

where the marked (red) points are the insertions of the projectors, which appear on the boundaries of the cylindrical surface, as indicated. (Readers interested in a more microscopically resolved interpretation of the above correspondence are invited to read Appendix A.) By viewing the matrix contractions as defining a Riemann surface via their ribbon graph, one quickly convinces oneself that the Wick contraction as shown gives rise to an annulus (or cylinder) type contribution. These are associated with Euclidean wormhole type geometries in the bulk [4, 26–28], as we will demonstrate directly in Section 4 below. In this way our formalism automatically produces connected correlations between different determinants, preventing their expectation values from factorizing. The two Wick contractions give one factor of $s^{-1}$ each, resulting in a total contribution $\sim s^{-2}$. Fourier transformed into the time domain, this gives precisely the linear in time behavior characteristic of the ramp. It has often been pointed out (see e.g. [29]) that the ramp physics is intrinsically non-perturbative, which is seen from our perspective from the fact that $s^{-2} \sim e^{-2S}$. The sigma-model approach transforms such – in principle –non-perturbative contributions into a simpler perturbative expansion (around the standard saddle).

However, as should be clear from the relation $Q = e^W \tau_3 e^{-W}$ this is but the leading order contribution in the expansion, and we now move on to higher-order examples. The next

simplest diagram comes from including in the action the term proportional to $\mathrm{STr}(B\tilde{B})^2$ in the expansion of $Q$, which gives us a contribution

$$\left\langle \mathrm{STr}(B\tilde{B}P^f)\mathrm{STr}(\tilde{B}BP^f)\mathrm{STr}((B\tilde{B})^2) \right\rangle$$

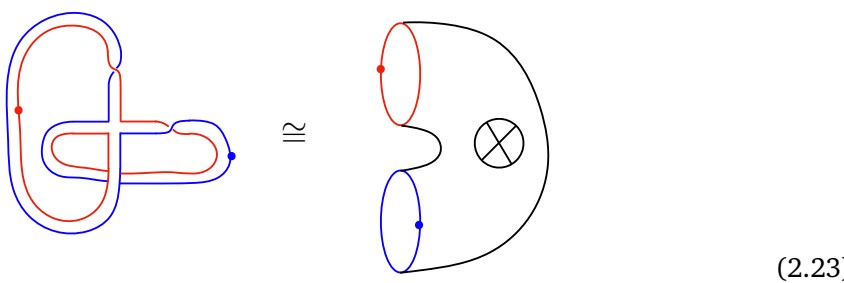

$$(2.23)$$

To avoid clutter, we have not shown the Wick contraction which, however, is reflected in the structure of the ribbon graph itself. We note that this contribution can only appear in a theory that has time reversal invariance, for example in the GOE symmetry class, which allows un-oriented ribbon graphs. The reader may convince herself that there is in fact no consistent set of arrows that can be drawn on the two closed loops of the ribbon graph above, so that each each is always anti-parallel to its neighboring edge (again, we refer to Appendix A for a more microscopically resolved discussion of this point.) Correspondingly, the bulk surface is non-orientable which is achieved by the crosscap insertion, as realized in bulk language in [26]. Finally, by expanding to yet higher order in the $B, \tilde{B}$ matrices, we can find higher-genus contributions, the first non-trivial one coming from expanding to order $\mathrm{str}((B\tilde{B})^4)$. This adds one handle as in

$$\left\langle \mathrm{STr}(B\tilde{B}P^f)\mathrm{STr}(\tilde{B}BP^f)\mathrm{STr}(((B\tilde{B})^2)^2) \right\rangle$$

$$(2.24)$$

In the unitary class, it actually turns out that a further genus-one contraction exists, which cancels precisely against the one shown here, but in other symmetry classes diagrams like the one above give non-vanishing contributions. Since the translation between ribbon graph and surface by eye can become a little complicated, one can proceed more abstractly. By tracing the loops of this diagram one sees that it is indeed orientable and that it has $F = 2$ faces, $V = 2$ vertices and $P = 4$ propagators, meaning that it has a single handle $H = 1$. This follows directly from the classic formula in toplogy $F - P + V = 2 - 2H$. The marked points corresponding to the $P^{\pm}$ projectors are again interpreted as brane boundaries. The bulk interpretation is a spacetime with two boundaries and non-trivial topology, i.e. a case where a baby universe has split off and re-fused in an intermediate channel.

The full perturbative expansion of the EFT proceeds by including higher and higher terms in the expansion of $Q$ in terms of the $B$ matrices. This results in successively higher-genus surfaces, each with two holes. The number of such holes is fixed to two, since we are computing a

spectral two-point function. It should be clear how this generalizes to higher-point functions. Note that the mathematical structure necessary for a connected correlator between different spectral determinants is a robust consequence of the symmetry-breaking scenario and its associated Goldstone physics. Notably this also gives a well-defined meaning to such correlations in a theory with fixed chaotic Hamiltonian and gives legitimacy to the appearance of Euclidean bulk wormholes in such cases.

Finally we should note that the expansion here has some similarity with the topological expansions of the JT matrix model [4] or indeed that of the older matrix models [3]. We will have more to say about this in Section 4.

### 2.6.1 Non-perturbative structure: second saddle and symmetry restoration

A major advantage of the EFT approach is that it provides access to the long time asymptotics of spectral correlations: for $s \lesssim 1$, corresponding to energies $\omega < \Delta$ or times $t > t_H$, the above perturbative expansion breaks down. Instead, the spectral correlation functions now probe the full Goldstone mode manifold. One may thus interpret the exploration of the full manifold in terms of the restoration of the causal symmetry. This symmetry restoration is natural from our earlier intuitive picture of the breaking of this symmetry: the symmetry *breaking* was reflected by the replacement of a discrete pole structure by a continuous cut at the level of the mean-field theory. However, once we probe finer structures $\omega \lesssim \Delta$, the theory is capable of reproducing the fact that the spectra of individual systems are discrete, and must therefore undo the effects of the symmetry breaking. It does so by allowing the Goldstone modes to explore the full coset space (see Fig. 1 for an illustration.)

Technically, the Goldstone mode matrix integrals Eq. (2.18) are simple enough to be doable in closed form for all symmetry classes [30]. However, for the present purposes it is not necessary to delve into the technicalities of these computations, a stationary phase analysis extended for the presence of the Altshuler-Andreev saddle, $Q_{AA}$, defined in the beginning of the section suffices for the present purposes. Referring the Reader to section 3 for the technical fine print, a repetition of the perturbative expansion, now around $Q_{AA}$ gives rise to the full structure

$$
R_2(s) = e^{s \times 0} \left( \underset{s^{-2}}{\vcenter{\hbox{}}} + \underset{s^{-4}}{\vcenter{\hbox{}}} + \cdots \vcenter{\hbox{}} \right)
$$
$$
+ e^{2is} \left( \underset{s^{-2}}{\vcenter{\hbox{}}}' + \underset{s^{-4}}{\vcenter{\hbox{}}}' + \cdots \vcenter{\hbox{}}' \right), \qquad (2.25)
$$

where for simplicity we included orientable diagrams/surfaces only. In the semiclassically exact unitary class, the expansion truncates after the first contributions, and we obtain the spectral two point function as

$$
R_2(s) = -\mathrm{Re}\, \frac{1}{2s^2}(1 - e^{-2is}) = -\frac{\sin^2 s}{s^2}. \qquad (2.26)
$$

As the above graphical representation suggests (see also Figure 1 for a description in terms of the target space geometry of the sigma model), the perturbative expansions around the two saddle points differ quantitatively, however they are organised in terms of topologically

equivalent surfaces, which we indicate by the primes after each contribution around the AA saddle. In summary then, the late-time behavior of chaotic quantum systems can understood semiclassically as an expansion around two saddles, with the perturbative series around each saddle organised into a topological series that can be interpreted as 'wormhole' and 'baby-universe' type contributions.

We will elaborate on this connection to bulk physics in Section 4 below, where we identify two-dimensional and three-dimensional bulk spacetimes corresponding to the universal $1/s^2$ singular diagram around the standard saddle above. It would be natural to interpret this as a holographic version of the periodic-orbit interpretation of spectral rigidity in quantum chaotic systems (see for example the book [31]).

In view of this relatively straightforward semiclassical interpretation of the expansion around the standard saddle, $Q_0$, one may ask if the expansion around the Altshuler-Andreev saddle, $Q_{AA}$, has a similar semiclassical interpretation, as well. This is particularly compelling in light of the fact that we can map one saddle-point contribution to the other using the Weyl symmetry as explained. The question is then, whether the topological expansion in $s^{-1}$ can also be interpreted term by term in terms of something like semiclassical orbits (and by extension, semiclassical bulk configurations). The answer is cautiously affirmative, although that interpretation is less well established and makes use of the so-called *Riemann-Siegel lookalike* [25]: First note that the semiclassical expansion in $s^{-1}$ is an asymptotic one. Its divergence for small $s$ reflects the exponentially growing number of long loops contributing to the trace of the resolvent, $W(z)$. On the other hand, the sum is the semiclassical approximation to something finite, the density of states, or the determinant of a random operator. Inspection of Eq. (2.26), and in particular of the negative sign multiplying the contribution from the AA saddle suggest an interpretation of orbits 'subtracted' from the full contents of these quantities (whatever that might be). These vague remarks can be made much more rigorous for certain proxies of chaotic systems such as $L$ dimensional random unitary matrices, $U$ [32]. In this case, the role of the spectral determinants is taken by $\det(1 - zU)$. These determinants are $L$-dimensional polynomials in $z$, where the secular coefficients, $A_n$, afford a representation as polynomials in $\text{Tr}(U^l)$, $l \leq n$, (the unitary analog of closed loops of length $l$.) Now, unitarity requires $A_n = \bar{A}_{N-n} A_N$. This key formula states that the contents of short orbits $0 < n \leq N/2$ determines that of the longest orbits, $N/2 \leq n \leq N$. Using this formula, and the aforementioned Weyl symmetry of the spectral determinant, the full spectral form factor of unitary random maps becomes accessible via perturbative expansion. Similar unitarity principles should apply to more complex chaotic systems and determine their non-perturbative spectral correlations. However, the mathematically sound implementation of these principles remains to be spelled out in practise.

# 3 EFT for matrix models

In this section, we discuss the EFT approach in the context of matrix models of arbitrary potential. Matrix models provide a valuable class of examples where the effective field theory of late-time chaos can be derived from first principles, serving as an explicit arena in which to illustrate each of the steps in Section 2.5 in full detail. At the same time, they represent duals to bulk theories [3, 4, 33], indicating that the symmetry breaking mechanism central to the present approach, too, has manifestations in the bulk. Before addressing this perspective in section 4, the two main goals of this section are 1) to explain in technical detail some of the steps in treating the $Q$-matrix theory and 2) to establish a set of examples in which the EFT of quantum chaos can be derived explicitly in order to gain some more intuition about its structure. In doing so we show in generality how to go from the $L \times L$ 'color-matrix' description

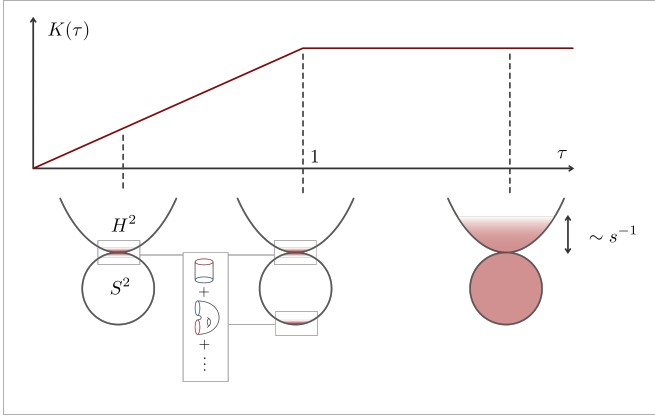

Figure 1: The generic late-time behavior described by the effective theory of quantum chaos. The top line shows the ramp and plateau behavior of a simple observable – the spectral form factor $K(\tau)$ – and the bottom line shows how the different qualitative regimes are obtained in the sigma model. The ramp behavior is initially well described by small perturbative corrections around the standard saddle (the north pole of the sphere in the sigma model geometry $H_2 \times S^2$). In this phase causal symmetry is spontaneously broken. Around the Heisenberg time $\tau = 1$, one must also include the non-perturbative contributions of the second saddle, as described in (2.25). At very late times the system explores the full Goldstone manifold $H_2 \times S^2$ and the causal symmetry is restored.

of the model to the graded flavor representation, where the rank of the matrix is $\mathcal{O}(1)$. (As we are solely mapping a high dimensional matrix integral to a single low dimensional one, the terminology 'field theory' may be a misnomer in the present setting. However, in view of the fact that the flavor integrals discussed here approximate higher dimensional field theories in the ergodic limit we keep using it.)

## 3.1 Invariant matrix models

An (invariant) matrix model is defined by a an ensemble of $L \times L$ random matrix Hamiltonians $H = \{H_{\mu\nu}\}$ governed by a probability distribution $P(H)dH = \exp(-V(H))dH$, where $V(H) = V(U^\dagger H U)$ is a unitarily invariant scalar function, and $dH$ a flat measure (possibly constrained to subsets of the hermitian matrices in cases where $H$ carries symmetries besides hermiticity). Assuming that $V(H) = \mathrm{Tr}(F(H))$ is expressed via the trace of a matrix function, we note the additional symmetry $V(H) = V(H^T)$ which we will use later in our construction.

Once again we focus on the spectral determinant (2.8), now averaged over the invariant distribution. We will demonstrate that this quantity can be exactly rewritten as a reduced integral over graded matrices, $A$, of much lower dimension $4 = 2 \times 2$. Here, one factor of two accounts for the different causality $\pm$ of the Green functions, and the second reflects the $\mathbb{Z}_2$ grading, i.e. the presence of commuting and anticommuting variables in $A$ required to generate determinants in the denominator and numerator, respectively. The $A$-matrix integral is form equivalent to the $H$-integral in that the integration is over the same distribution $P(A)dA = \exp(-V(A))dA$ and a structurally identical integrand. However, it is much easier to handle, especially when it comes to the description of correlations at 'microscopic' scales of the order of the level spacing.

## 3.2 Construction of the theory

As explained in section 2.5 above, our starting point is the Gaussian integral representation, Eq. (2.11), of the spectral determinant Eq. (2.8), now integrated over the invariant ensemble

$$\langle \mathcal{Z}(\hat{z}) \rangle = \int dH \int d(\bar{\psi}, \psi) \, e^{-V(H)+i\bar{\psi}(\hat{z}-H)\psi} \,. \tag{3.1}$$

Let us write $\psi = (\phi^+, \phi^-, \eta^+, \eta^-)^T$, $\bar{\psi} = (\bar{\phi}^+, -\bar{\phi}^-, \bar{\eta}^+, \bar{\eta}^-)$, so that $\phi^\pm$ denote c-number integration variables, $\bar{\phi}^-$ their conjugates (note the minus sign in front of $\bar{\phi}^-$ required by convergence), $\eta^\pm, \bar{\eta}^\pm$ are independent Grassmann variables, and we have made the $H$-average explicit. We arrange the energies as a matrix,

$$\hat{z} = E + \hat{\omega} + i\delta\tau_3 \,, \qquad \hat{\omega} = \text{diag}(\omega_1, \omega_2, \omega_3, \omega_4) \,, \tag{3.2}$$

where the energy arguments have been split into $E\mathbb{1}_{4\times 4}$, and the matrix of energy differences $\omega$, and we assume $\sum_i \omega_i = 0$ without loss of generality. The integral above is set up such that integration over the commuting (anti-commuting) variables produces the determinants in the denominator (numerator) entering the spectral determinant. Note that the integration vectors $\psi = \{\psi_\mu^a\}$ and $\bar{\psi} = \{\bar{\psi}_\mu^a\}$ live in a tensor product space, $\mathcal{H}_C \otimes \mathcal{H}_F$, where $\mathcal{H}_C$ is the $L$-dimensional 'color' representation space of the Hamiltonian and $\mathcal{H}_F$ the four dimensional graded 'flavor' space of $\psi$'s internal degrees of freedom. Defined in this way, $\psi$ carries two natural group representations. The first is a unitary representation,

$$\psi \to U\psi \,, \qquad \text{where} \qquad U = \{U_{\mu\nu}\} \in \text{U}(L) \,. \tag{3.3}$$

Due to the invariance of the distribution, $P(H)dH = P(UHU^{-1})dH$, this defines an exact symmetry of the integral (3.1). The second is a representation under graded flavor matrices,

$$\psi \to T\psi \,, \bar{\psi} \to \bar{\psi}T^{-1} \,, \qquad T = \{T^{ab}\} \in \text{GL}(2|2) \,. \tag{3.4}$$

Invariance under this transformation is weakly broken by differences between the frequency arguments, $\omega_i$, and infinitesimally by the causal increment, $i\delta$. We will see that these two symmetries and their fate under $H$-averaging essentially determine the matrix integral.

### 3.2.1 The color flavor map

To prepare the average over $P(H)$, we define the two matrix structures,

$$\Xi \equiv \bar{\psi}\psi = \{\bar{\psi}_\mu^a \psi_\nu^a\} \,, \qquad \Pi \equiv \psi\bar{\psi} = \{\psi_\mu^a \bar{\psi}_\mu^b\} \,, \tag{3.5}$$

where the first is an $L \times L$ color matrix transforming as a flavor singlet, and the second a $4 \times 4$ graded flavor matrix transforming as a color singlet. The $H$-dependent term in the integrand can be written in the form $\bar{\psi}H\psi = \text{Tr}_c(H^T\Xi)$, and the average over $H$ yields

$$\left\langle e^{i\bar{\psi}H\psi} \right\rangle_H = \left\langle e^{i\,\text{Tr}(H^T\Xi)} \right\rangle_H = \left\langle e^{i\,\text{Tr}(H\Xi)} \right\rangle_H \equiv G(\Xi) \,, \tag{3.6}$$

where in the third equality we used the invariance of the distribution under $H \to H^T$, and in the final one defined $G(\Xi)$ as the generating function of the distribution $P(H)$. For the purposes of the present construction, there is no need to know the function $G$ explicitly. However, the above mentioned unitary invariance (3.3) implies $G(U^{-1}\Xi U) = G(\Xi)$. In symbolic notation, we assume this invariance condition to be realized as $G(\Xi) = G([\text{Tr}_c(\Xi^n)]^m)$, i.e. via dependence of $G$ on arbitrary powers of traces of powers of $\Xi$. The key relation now coming into play is

$$\text{Tr}_c(\Xi^n) = \text{STr}_f(\Pi^n) \,. \tag{3.7}$$

The identity is proven by the cyclic exchanges of the $\psi_\mu^a$ fields in writing out the color trace explicitly.

As a consequence of this relation, we have $G(\Xi) = G(\Pi)$, where in a slight abuse of notation we denote the function $G(\Pi) = G([\mathrm{STr}_f(\Pi^n)]^m)$ by the same symbol, $G$. We may now pass back from the generating function to a distribution as

$$G(\Delta) \equiv \left\langle e^{i\,\mathrm{STr}_f(A\Pi)} \right\rangle_A,$$

where we introduce the average

$$\langle \dots \rangle_A = \int dA\, P(A)(\dots), \tag{3.8}$$

over the four-dimensional graded matrix, $A$ with respect to the flat measure $dA$, meaning an independent integration over all commuting and anticommuting variables. Here, $P(A)$ is the flavor matrix distribution defined by the generating function $G(\Pi)$. Since $G$ did not change its form in passing from $\Xi$ to $\Pi$, the same it true for the distribution $P$ in passing from $H$ to $A$. Note that for the simple example of a Gaussian matrix ensemble,

$$V(H) = \frac{L}{g^2}\mathrm{Tr}(H^2), \tag{3.9}$$

the equivalence

can be verified by elementary manipulation of Gaussian integrals. Substituting the result back into the starting expression, we obtain

$$\langle \mathcal{Z}(\hat{z}) \rangle = \int dA \int d(\bar{\psi}, \psi)\, e^{-V(A) + i\bar{\psi}(z - A)\psi}, \tag{3.10}$$

an expression identical to Eq. (3.2), except that the integral over the $L \times L$ Hamiltonian $H$ is replaced by one over the much smaller flavor matrix $A$. The advantage of these manipulations become obvious once we integrate over $\psi$

$$\langle \mathcal{Z}(\hat{z}) \rangle = \left\langle \mathrm{sdet}(\hat{z} - A)^L \right\rangle_A = \left\langle e^{-L\,\mathrm{STr}\ln(\hat{z} - A)} \right\rangle_A, \tag{3.11}$$

to obtain a (super-)determinant[6] raised to the $L$-th power due to fact that the integrand is a singlet with respect to its Hilbert-space indices. Making the averaging procedure explicit, we obtain a dual representation of the integral

$$\langle \mathcal{Z}(\hat{z}) \rangle = \int dA\, e^{-V(A) - L\,\mathrm{STr}\ln(z - A)}, \tag{3.12}$$

now formulated in terms of the low dimensional flavor matrices. From now on, most traces will be over flavor space, and we omit the corresponding subscript.

Building on this representation, we may now retrace the individual steps outlined in section 2.5, notably the GL(2|2) symmetry, and its spontaneous broking down to GL(1|1)×GL(1|1). We will go quickly through the derivation of the EFT building on this symmetry breaking scenario, but emphasize certain technical details that were left out in the more conceptual treatment above.

---

[6]The super-determinant of a graded matrices is defined as $\mathrm{sdet}\left(\begin{smallmatrix} a & \rho \\ \tau & b \end{smallmatrix}\right) = \det a / \det(d - \tau a^{-1}\rho)$.

### 3.2.2 Deriving the effective sigma model

The presence of the large pre-factor $L$ in the exponent motivates a stationary phase analysis of the integral. Variation of the action yields the saddle point equation

$$\delta_A S(\bar{A}) = V'(\bar{A}) - L\frac{1}{\hat{z} - \bar{A}} = 0. \tag{3.13}$$

To get the spectral density, we differentiate the spectral determinant once in energy to obtain,

$$
\begin{aligned}
-\partial_{\omega_1}\big|_{\omega=0} \langle \mathcal{Z}(\hat{z}) \rangle &= \left\langle \mathrm{Tr}_c\left(\frac{1}{E^+ - H}\right) \right\rangle_H \\
&\simeq \left(\frac{N}{E + i\delta\tau_3 - \bar{A}}\right)_{11},
\end{aligned} \tag{3.14}
$$

where in the second line we have evaluated the expression at the stationary point. The approximate equality sign indicates that the relations holds up to $1/L$ corrections. Accordingly, the average resolvent is straightforwardly obtained by solving the stationary phase equation at $\omega = 0$. Specifically, the average spectral density, $\rho(E)$, follows from taking the imaginary part of (3.14) evaluated at the saddle point. To illustrate this point, consider the case of the Gaussian matrix potential, $V(A) = \frac{L}{g^2}\mathrm{Tr}(A^2)$, for which the variational equation at $\omega = 0$ takes the form

$$g^2\bar{A} - (E + i\delta\tau_3 - \bar{A})^{-1} = 0.$$

Reflecting the rotational symmetry of the action, this equation affords solution in terms of diagonal matrices $\bar{A}$. It is a quadratic equation, individually for each of the diagonal matrix elements, and we need to pick one of two solutions — this is where the spontaneous symmetry breaking happens. Specifically, for $|E| < 2g$, we have the solutions $\bar{A}(E) = E/2 \pm i\gamma(E)$, and the 'natural' of these is

$$\bar{A}(E) = \frac{E}{2} + i\gamma(E)\tau_3, \qquad \text{where} \qquad \gamma(E) = \sqrt{g^2 - \left(\frac{E}{2}\right)^2}. \tag{3.15}$$

Physically, this solution is natural in that the sign of the imaginary part $i\gamma$ is dictated by the infinitesimal $i\delta$. We interpret this as shifting of a pole into the complex plane reflecting the smearing of a discrete pole structure into a cut at mean field level[7]. Substitution of this variational solution into Eq. (2.9) leads to the famous semicircular density of states

$$\rho(E) = -\frac{1}{\pi}\mathrm{Im}\,\mathrm{Tr}_c(G^+(E)) = \frac{L}{\pi g}\left(1 - \left(\frac{E}{2g}\right)^2\right)^{1/2}. \tag{3.16}$$

Returning to the case of an arbitrary invariant potential, suppose that a solution to the saddle-point equation of the form

$$\bar{A}(E) \equiv \epsilon(E) + i\gamma(E)\tau_3, \tag{3.17}$$

---

[7]In the bosonic sector of indices diagonal indices $|a| = 1$, this sign choice actually is obligatory. The reason is that the saddle points with their finite imaginary part must be reached by deformation of the real integration contours defined by the eigenvalues of the Hermitian integration variables $A^{ab}$, $|a| = |b| = 1$. Under the logarithm, these variables appear infinitesimally shifted into the upper/lower complex half plane, depending on the sign $\pm i\delta$. Reaching the 'wrong' saddle points would require passage through the cut of the logarithm and cause a a divergence in the integral. (This is best seen in the representation, $\exp(-\ln(E \pm i\delta + x)) = 1/(E \pm i\delta + x)$.) In the fermionic sector, no such problem exists, and either saddle point is reachable.

with real $\epsilon, \gamma$ has been found. We then build on the spontaneous breaking of the causal symmetry breaking reflected by the $\pm\gamma(E)$ term, and turn to step 3 in constructing the EFT which instructs us to define the matrix $Q$

$$\bar{A} \to T\bar{A}T^{-1} \equiv \epsilon(E) + i\gamma(E)Q, \qquad Q \equiv T\tau_3 T^{-1}, \tag{3.18}$$

parametrizing the Goldstone manifold (2.13) of stationary solutions, where $T$ has been defined in (2.14). Moving along the general program we can now write down the effective action using this object.

*Symmetry breaking and effective action:* we begin with the substitution of Eq. (3.18) into the action of Eq. (3.12). The GL(2|2) invariance of the potential term, $V(A) = V(TAT^{-1})$, implies the decoupling of the former from the Goldstone mode integral. (This is an elegant way of seeing why the singular Goldstone mode fluctuations are oblivious of the detailed form of the invariant distribution. Their job solely is to determine the average spectral density via the above $\gamma(E)$.) The action thus reduces to

$$S[Q] = L \operatorname{STr} \ln\left(\tilde{E} + \hat{z} + i\gamma Q\right),$$

where we redefined $\hat{z} \to \hat{z} - E$ to single out the symmetry breaking parameters and absorbed the real part of the stationary point solution into a redefined energy parameter $\tilde{E} = E - \epsilon(E)$. Using the cyclic invariance of the trace, $S[Q] = L \operatorname{STr} \ln\left(\tilde{E} + T^{-1}\hat{z}T + i\gamma\tau_3\right)$, we couple the Goldstone mode fluctuations to the explicit symmetry breaking, $z$. In a final step, we expand to first order in $z/\gamma$ and use the saddle point property $(\tilde{E} + i\gamma\tau_3)^{-1} = (E - \bar{A})^{-1} = \frac{i\pi}{L}\rho\tau_3 + \dots$, where $(\dots)$ are contributions proportional to the unit matrix to obtain the effective action

$$S[Q] = i\pi\rho \operatorname{STr}(\hat{z}Q). \tag{3.19}$$

It should come as no surprise that this coincides with the ergodic limit (2.17) of the EFT for extended systems. We are now in a position to explore the physics of the Goldstone modes in the setting of a model which does not contain any 'spatially fluctuating' modes. This goes over the same physics as as visualized in 2.5 using ribbon graphs, but we wish to take the opportunity here to give a careful derivation of all the prefactors and signs that we had previously glossed over.

## 3.3 The integral over the Goldstone manifold

In this section, we turn to the details of the $Q$-matrix integration over the identical actions Eq. (3.19), or Eq. (2.17). Their equality implies that the results derived here equally apply to the matrix model and to the physics of extended systems below their Thouless energy.

### 3.3.1 One-point functions: the spectral density

In general, the method of choice for doing the $Q$ integration depends on both, the magnitude of $\omega$, and the specific observable to be computed, reflected in a judicious choice of the sources for advanced and or retarded correlation functions. Note that in our previous treatment the source structure was reflected in the way the projectors $P^{\pm}$ appeared in correlators such as (2.22). In the following, we address three instances of interesting source insertions and $\omega$-ranges, and along the way compute the ramp-plateau profile of GUE spectral statistics.

The evaluation of the integral is particularly easy in cases where the integrand possesses a fully unbroken supersymmetry, i.e. invariance under a supersubgroup GL(1|1). Under these

circumstances, a theorem due to Efetov and Wegner states that the integral collapses to its value at the 'coset origin',[8]

$$\int dQ e^{-S[Q]} \stackrel{\text{susy}}{=} e^{-S[\tau_3]}. \tag{3.20}$$

As an example, consider the case of absent sources, and degenerate energy arguments $\hat{z} = i\delta\tau_3$. In this case, the spectral determinants cancel out, and $\langle \mathcal{Z}(\hat{z}) \rangle = 1$. This is confirmed as

$$\left\langle \mathcal{Z}_{(4)}(0,0,0,0) \right\rangle = e^{-\delta\pi\rho \, \text{STr}(\tau_3)} = 1, \tag{3.21}$$

where the action reads more explicitly $\text{STr}(\tau_3) := \text{STr}(\tau_3 \otimes 1_{2\times 2}) = 1$, by supersymmetry. Differentiating once with respect to sources, as in (3.14), confirms the identification of the prefactor of the action as the density of states $\rho$, which can be evaluated as demonstrated in 3.2.2 above.

### 3.3.2 Perturbative spectral correlations: the ramp

We next apply the formalism to the computation of *spectral fluctuations*, as described by the cumulative spectral two-point function (2.10). We rewrite (2.10) slightly, using the definition of the mean level spacing as

$$
\begin{aligned}
R_2(\epsilon) &= \Delta^2 \left\langle \rho\left(E + \frac{\epsilon}{2}\right)\rho\left(E - \frac{\epsilon}{2}\right)\right\rangle_c \\
&= \frac{\Delta^2}{2\pi^2} \text{Re} \left\langle \text{Tr}(G^+(E + \tfrac{\epsilon}{2}) \text{Tr}(G^-(E - \tfrac{\epsilon}{2}) \right\rangle_c,
\end{aligned} \tag{3.22}
$$

where we used that the connected average $\langle G^\pm G^\pm \rangle_c \equiv \langle G^\pm G^\pm \rangle - \langle G^\pm \rangle \langle G^\pm \rangle$ of Green functions of coinciding causality vanishes. Inspection of the spectral determinants shows that the average of Green functions in this expression is obtained as

$$R_2(\epsilon) = \frac{\Delta^2}{2\pi^2} \text{Re} \, \partial^2_{\omega^{f+}, \omega^{f-}} \left\langle \mathcal{Z}(\hat{z}) \right\rangle_c, \tag{3.23}$$

where the derivative is evaluated at the configuration $\omega = \frac{\epsilon}{2}\tau_3$, and the subtraction of a 'disconnected' contribution to the functional integral, indicated by the subscript $c$, implements the cumulative average. Doing the derivative in the representation (2.18), we arrive at the representation

$$
\begin{aligned}
R_2(\epsilon) &= -\frac{1}{2} \text{Re} \left\langle \text{STr}(Q(P^+ \otimes P^f)) \, \text{STr}(Q(P^- \otimes P^f)) \right\rangle_{Q,c}, \\
\langle \dots \rangle_Q &\equiv \int dQ \, e^{-i\frac{s}{2} \text{STr}(Q\tau_3)}(\dots),
\end{aligned} \tag{3.24}
$$

where $P^s, (s = \pm)$ projects on the advanced and retarded causal sector, respectively, while $P^{f,b}$ project on the fermionic and bosonic sector, respectively, and we represented the action (after source differentiation) in the dimensionless units Eq. (2.3). This is the matrix theory whose topological expansion, ordered by powers of $s$, we discussed in Section 2.5. We now carry out the integral explicitly, thus fixing the coefficients of each term in that expansion. In a first

---

[8]The rational behind this collapse is that in the case of a supersymmetry, we have a conflict of interests: due to the invariance, the integrand does not depend on the Grassmann valued generators, $\eta$, of the symmetry, and the integration $\int d\eta = 0$ suggests a vanishing integral. On the other hand, it also does not depend on the non-compact bosonic, $x$, variable of the symmetry subgroup, and the integral $\int dx = \infty$ suggests a divergence. The solomonic resolution of the $0 \times \infty$ conflict is a collapse of the integral to configurations where all generators vanish. For a pedestrian discussion of the details, we refer to Ref. [31].

step, we concentrate on large energy offsets, $\epsilon \gg \Delta$ or $s \gg 1$. In this case, the action of the matrix integral is strongly oscillatory, and Goldstone mode fluctuations are confined to small neighborhoods $\sim s^{-1}$ of the stationary points on the saddle point manifold. These residual fluctuations are conveniently described in an (exponential) parameterization,

$$Q = T\tau_3 T^{-1}, \qquad T = \exp W, \qquad W = \begin{pmatrix} & B \\ \tilde{B} & \end{pmatrix}, \tag{3.25}$$

where the block structure implements the anti-commutativity $[W, \tau_3]_+$ of the generators with the coset origin, and the $2 \times 2$ super-matrices

$$B = \begin{pmatrix} z & \mu \\ \nu & w \end{pmatrix}, \qquad \tilde{B} = \begin{pmatrix} \bar{z} & \bar{\nu} \\ \bar{\mu} & -\bar{w} \end{pmatrix}, \tag{3.26}$$

contain the two complex commuting $(z, w)$ and four Grassmann $(\mu, \ldots, \bar{\nu})$ integration variables of the model, parametrizing the most general Goldstone fluctuation. As we stated earlier in more general terms, we can see explicitly that in the bosonic sector, $B^{\mathrm{bb}}$ the variable $z$ spans a hyperboloid with radial coordinate $|z|$ and in the fermionic sector $B^{\mathrm{ff}}$ the variable $w$ a sphere, $Q^{\mathrm{ff}} = n^i \tau_i$, with $|n| = 1$, where $|w| = \pi/2$ represents a rotation from the north pole, $n = (0, 0, 1)^T$, to the south pole, $n = (0, 0, -1)^T$, i.e. the transformation $T_W$ mapping the standard onto the AA saddle. Let us now substitute an expansion to quadratic order in fluctuations

$$Q = \tau_3(1 - 2W + 2W^2 + \ldots),$$

into Eq. (3.24), we obtain up to quadratic order the expression,

$$R_2(\epsilon) = -2\,\mathrm{Re}\left\langle \mathrm{STr}(B\tilde{B}P^{\mathrm{f}})\,\mathrm{STr}(\tilde{B}BP^{\mathrm{f}}) \right\rangle,$$

where the $B$-average is defined in Eq. (2.21), and the disconnected pure saddle point contribution, $B = 0$, does not contribute to the cumulative average. Note that we have now recovered in detail the structure we had described in a more qualitative setting in Eqs. (2.22) – (2.24) above. Doing the final Gaussian integral[9], we obtain

$$R_2(s) \simeq -\frac{1}{2s^2}, \tag{3.27}$$

corresponding to the ramp, $K(\tau) = \tau$, upon Fourier transformation to dimensionless time. The strategy for refining this result beyond the leading order in the parameter $s^{-1} \ll 1$ is evident: expansion of the action in the generators $B$ leads to terms $\sim s\,\mathrm{STr}(B\tilde{B})^n$, which after integration contribute as $s^{1-n}$. However, in the unitary symmetry class, it turns out that the prefactors of all these contributions cancel out order-by-order in the $s^{-1}$ expansion, and that (3.27) does not change in perturbation theory.

### 3.3.3 Non-perturbative correlations: Weyl symmetry and the plateau

For $s < 1$, the stationary phase approach to the Goldstone mode is no longer parametrically controlled. At the same time, the absence of perturbative corrections to the approximation Eq. (3.27) hints at a 'semiclassically exact' integral. Heuristically, the underlying mechanism can be understood by inspection of the fermionic sector of the theory: with $Q^{\mathrm{ff}} = n^i \tau_i$, the action $\mathrm{tr}(Q^{\mathrm{ff}}) = 2n_3$ is proportional to the height function $n_3 = \cos(\theta)$ on the sphere, and

---

[9]This can be done by brute force, or using the matrix version of Wick's theorem, $\langle \mathrm{STr}(BX\tilde{B}Y) \rangle = \frac{1}{is}\mathrm{STr}(X)\mathrm{STr}(Y)$, $\langle \mathrm{STr}(BX)\mathrm{STr}(\tilde{B}Y) \rangle = \frac{1}{is}\mathrm{STr}(XY)$.

$\int dQ^{\text{ff}}$ an integration over the canonical measure. The integral thus affords an interpretation as partition sum of a spin precessing in a fictitious magnetic field of strength $s$. This partition sum is a classic example of the semiclassical exactness principle [34], and it turns out that this feature carries over to its supersymmetric extension[10].

However, the semiclassical exactness principle requires us to account for all stationary points of the integrand. We already mentioned the stationarity condition, $[Q, \tau_3] = 0$, which besides by the standard saddle $Q = Q_0 = \tau_3$ is solved by the AA saddle $Q = T_W Q_0 T_W^{-1} = \tau_3 \otimes \sigma_3^{\text{bf}}$, with stationary action $S(Q_{\text{AA}}) = is\,\text{STr}(\tau_3 \otimes \sigma_3^{\text{bf}}) = 2is$. Referring for details of the Gaussian integration around the AA saddle point to the original reference [35], we note that it produces the same factor as in Eq. (3.27), but with inverted sign. Adding the two terms, we obtain the full result Eq. (2.26). For completeness, we mention that the full Goldstone mode integrals required to obtain the correlation functions in other ensembles are doable in closed form by introducing coset space 'polar coordinates' and using the high degree of rotational invariance of the action. For the technical details, interested readers are referred to to Ref. [30].

# 4 Causal symmetry breaking in the bulk

We will now describe the emergence of the sigma-model from bulk considerations. The main ingredient to develop a bulk understanding is, once again, the determinant operator

$$\det(E^{\pm} - H) = e^{\text{Tr}\log(E^{\pm} - H)}. \tag{4.1}$$

This object should be thought of as inserting a D-brane in the bulk at position $E$ (see for example [36, 37]), and we distinguish here between an advanced brane and a retarded brane, depending on the sign of the (infinitesimal) imaginary part of the energy, as indicated. The Hermitian matrix $H$ – the Hamiltonian from the point of view of our discussion of quantum chaos above – corresponds to the presence of an additional stack of D-branes, one for each dimension of the many-body Hilbert space. One way to see this is by noting that the determinant operator can be obtained as the exponential of the loop operator

$$\mathcal{W}(E) = \text{Tr}\log(E - H), \tag{4.2}$$

which acts to insert a boundary in an open string wordsheet with boundary condition such that the string now ends on the '$E$-brane'. Upon exponentiating to form the determinant, the ascending powers of the $\text{Tr}\log E - H$ insertion correspond to open string world sheets with an increasing number of boundaries with the right combinatorics to count all possible ways the string can end on the D-brane [36]. To summarise then, $H$ is an $L \times L$ matrix corresponding to the presence of a (large) stack of $L$ D-branes, while $E$ corresponds to the position of a single additional "probe" D-brane. Given that these additional branes are added in order to construct ratios of spectral determinants as in (2.8), we sometimes refer to these objects as "spectral branes". We will return to more concrete microscopic realizations of such objects below in the context of minimal string theory, where we can take the $L$ branes to be of ZZ type and the $\mathcal{O}(1)$ spectral branes to be of FZZT type [33, 38, 39]. The physics we want to focus

---

[10]Recall that an integral is semiclassically exact if it assumes the form of a partition sum over an effective Hamiltonian, all whose trajectories are periodic and have equal revolution time. Both conditions are met in the present context: the Goldstone mode manifold is symplectic with $\omega \equiv \text{STr}(QdQ \wedge dQ)$ as defining two form. The integration extends over the symplectic measure, and the integrand contains the exponentiated Hamiltonian $H \equiv -is\,\text{STr}(Q\tau_3)$. For a given initial point. $Q$, the trajectories $Q(t)$ satisfy the condition $\iota_{\dot{Q}}\omega = 2\,\text{STr}(Q\dot{Q}dQ) = dH$, i.e. the tangent to the flow is a Hamiltonian vector field. A straightforward computation shows that for our present 'spin in a magnetic field of strength $s$' Hamiltonian, this condition is met by $Q(t) = e^{ist\tau_3}Qe^{-ist\tau_3}$ 'spin precessing at constant frequency $s$'. All trajectories are closed and have equal revolution time $2\pi/s$.

on is that of open strings stretching between these different types of branes. Let us introduce these degrees of freedom by writing the determinant operator explicitly as

$$\det(E - H) = \int d(\bar{\eta}, \eta) \exp(\bar{\eta}(E - H)\eta) , \tag{4.3}$$

where $\bar{\eta}, \eta$ are our usual $L-$component Grassmann vectors, now identified with fermionic open-string modes stretching between a single brane parametrized by $E$, and the $L$ branes in the stack parameterised by $H$. A second kind of excitation is given by bosonic strings stretching between the branes, in which case we exponentiate the *inverse* determinant operator as

$$\frac{1}{\det(E^{\pm} - H)} = \int d(\bar{\phi}, \phi) \exp\left(\pm i \bar{\phi}(E' \pm i\varepsilon - H)\phi\right) , \tag{4.4}$$

where we have been careful about adding an imaginary part ($\varepsilon > 0$) to ensure convergence of the Gaussian integral, in the same way as in Section 2.5 above. Such inverse brane determinants have been considered in the past, and have been called anti-branes or, perhaps more appropriately as ghost branes [40]. As an alternative to using ghosts, we might employ $2R$ replica branes of the 'normal' FZZT type. However, except for the unitary class, replicas are ill suited to the description of the doubly non-perturbative limit [41] which is why we generally prefer to work with ghosts (See Appendix B for a discussion of replicas vs. supersymmetry from the matrix theory perspective.)

It is thus clear how to interpret a generating functional of the type $\mathcal{Z}(\hat{z})$ (see Eq. (2.8)) in the bulk. We then have again the object (2.8)), but now with an explicit *physical realization* of the modes $\psi = (\phi^{+}, \phi^{-}, \eta^{+}, \eta^{-})$ which we had previously introduced as auxiliary objects. As we have seen, the ratio of determinants defining the generating functions $\mathcal{Z}$ is the starting point to extract universal late-time chaos from symmetry considerations alone. Given our interpretation of the $\eta$ and $\phi$ variables used to exponentiate the D-brane operator as open string modes. These string modes have Chan-Paton factors $\psi_{\mu}^{a}$, $\mu = 1, 2 \dots L$, $a = 1, 2 \dots n$ which allow them to end on any of the $L$ D-branes making up the 'sea' and/or on any of the $n$ probe branes. Note that the present approach is supersymmetric by design, with a supergroup acting upon the Chan-Paton degrees of freedom (see for example [40] for a general description of such objects). Both the replica and the SUSY perspective reveal that the bulk manifestation of the Goldstone modes of the chaotic sigma model are effective bound states of strings

$$\Pi^{ab} = \sum_{\mu=1}^{L} \psi_{\mu}^{a} \bar{\psi}_{\mu}^{b} ,$$

resulting from integrating out over the 'sea' degrees of freedom associated with the stack of $L$ D-branes labelled by the index $a$. This projects onto singlets under this 'color' group, leaving the effective strings to transform in with their $a, b$ indices in the adjoint of the 'flavor' group. For the computation of spectral correlations, for example of pair correlations (2.6), the causal symmetry breaking mechanism applies and we may effectively concentrate on the light sector of modes contained in $\Pi$, in other words the Goldstone sector $Q$.

This bulk interpretation of the chaotic sigma model is illustrated in Figure 2.

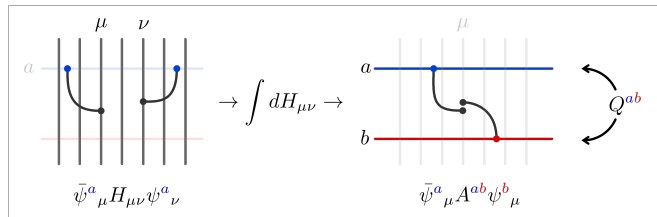

Figure 2: Bulk sigma model: we start from the configuration comprising 'sea branes', $\mu = 1, \dots L$, and spectral branes, $a = 1, \dots 4$. As a microscopic realization, for example in the context of minimal string theory, we can take the $L$ 'sea branes' to be a stack of $L$ coincident ZZ branes and the spectral branes to be FZZT branes. In the corresponding matrix theory, pairs of strings connecting sea branes $\mu$ and $\nu$, respectively, to the same spectral brane, $a$, are represented by the bilinears $\bar{\psi}^a_\mu H_{\mu\nu} \psi^a_\nu$, with Chan-Paton factors $\psi^a_\mu$. Integration over the stack of sea branes leads to a dual picture, with 'string bound states' $\bar{\psi}^a_\mu A^{a,b} \psi^b_\mu$ correlating spectral branes, $a, b$, via an induced bulk geometry with each other. The previously exact symmetry between different $a$'s (for spectral branes at coinciding energy-coordinates, $E$) is spontaneously broken and leads to the emergence of the $Q^{ab}$ pseudo Goldstone degrees of freedom. The reader may note that this is a microscopic description of an open/closed duality in the presence of 'flavor' branes [42] giving rise to an additional open string sector (the FZZT-ZZ strings). In section 4.2 we show how to extract the Goldstone contributions from a world-sheet calculation of the FZZT-ZZ strings.

## 4.1 Causal symmetry breaking in minimal string theory

We now go through the exercise of describing the bulk picture of the EFT of quantum chaos in minimal string theory. This allows us to make direct contact with the matrix-model techniques introduced above and then compare them to the bulk 2D gravity or 'continuum worldsheet' perspective. Referring to Refs. [33, 43, 44] for reviews, we note that minimal string theory is defined by coupling 2D Liouville theory to a $(q, p)$ minimal model matter CFT (for $q, p$ two relatively prime integers). For concreteness we will be only interested in the cases $(2, p)$, which have a dual description as one-matrix models with invariant potentials depending on the value of $p = 2m - 1, (m \in \mathbb{Z})$. It turns out that minimal string theories contain D-branes of the type we can use to implement the spectral determinant construction at the heart of our analysis. These D-branes can be viewed as CFT boundary states, tensoring a Liouville boundary state, [38, 45], with a matter boundary state, or alternatively as boundary conditions on the worldsheet theory. Reference [44] contains an extensive review of the relevant constructions[11] from a perspective pertinent to this work. In the following we will work both from the matrix-model perspective as well as from the worldsheet perspective, the latter taking the role of bulk spacetime. The former description will be very familiar from our Section 3, while the latter will recover individual universal contributions from the 2D gravity perspective.

---

[11]We also refer the interested Reader to the excellent review [3] for more information. A brief overview from a modern perspective is given in [33].

### 4.1.1 Double scaling to the spectral edge

One way of defining $(2, q)$ minimal string theory is via a double-scaled limit of a random matrix ensemble of a single matrix which we may think of as the Hamiltonian $H$, so that

$$\langle \mathcal{Z}(\hat{z}) \rangle = \int dH e^{-V(H)} \mathcal{Z}(\hat{z}), \tag{4.5}$$

falling squarely into the class of models (3.1) studied in Section 3. However, the present discussion requires an extra twist, we need to zoom into the vicinity of the spectral edge. To understand what this means in the present context, first consider the generating functional $\mathcal{Z}_{(2)}$ Eq. (2.9) of the spectral density with derivative taken at $z = E \pm i0$. Depending on the choice of $V(H)$, there will be a subset of energies $E$ with finite spectral density — technically, the range of energies with symmetry broken stationary field solutions. Symbolically denoting the width of this spectral support by $a$, and its minimum energy by $E_0$, we consider the 'double scaling limit', of a large number of levels, $L \to \infty$, for separations off the band edge $(E - E_0)/a \to 0$. In the following we consider the simplest case, namely $(2, 1)$ minimal string theory, whose invariant matrix potential is Gaussian, Eq. (3.9), to demonstrate how this limit leads to a variant of the Kontsevich matrix model [46]. However, the same method is applicable to other potentials describing theories in the $(2, q)$ family. For the discussion of the equivalent continuum worldsheet approach we refer to section 4.2 below.

After integrating out the variables $\psi, \bar{\psi}$ (now interpreted as open string degrees of freedom) we obtain the action (3.12), which we repeat here for convenience:

$$\langle \mathcal{Z}(\hat{z}) \rangle = \int dA e^{-V(A) - L \operatorname{STr} \ln(\hat{z} + A)}, \tag{4.6}$$

with $V(A) = \frac{L}{g^2} \operatorname{STr}(A^2)$, for the $(2, 1)$ variant. The solution $\bar{A}$ of the stationary equations is given by Eq. (3.15) with associated spectral density Eq. (3.16), indicating that $E_0 = -\sqrt{2}g$ defines the edge of a spectrum of width $a = 2\sqrt{2}g$. We now consider energies close to the band edge, $z = -E_0 + \zeta$. For fixed $\zeta$, we scale $L \to \infty$ along with $g \to \infty$ such that there is still a macroscopically large number of levels in the interval $[-E_0, -E_0 + \zeta]$. In this limit, Eq. (3.17) reduces to

$$\bar{A}(\zeta) = \frac{-\sqrt{2}g + \zeta}{2} + i \tau_3 \frac{g^{1/2}}{2^{1/4}} \sqrt{\zeta}, \tag{4.7}$$

with associated mean field spectral density

$$\rho(\zeta) = \frac{e^{S_0}}{\pi} \sqrt{\zeta}, \qquad e^{S_0} \equiv \frac{2^{3/4} L}{g^{3/2}}. \tag{4.8}$$

We obtain a two-dimensional flavor matrix model describing the fine structure of edge of single level spacings by expansion around the symmetry unbroken solution right at the edge, $\bar{A}(0) = \frac{g}{\sqrt{2}}$. Defining $A = \bar{A}(0) + \frac{\sqrt{g}}{2^{1/4}} a$ where the scaling factor upfront the fluctuation matrix $a$ is introduced for convenience, and expanding to leading order in $a$ and $\zeta$, we obtain

$$\langle \mathcal{Z}(\hat{z}) \rangle \simeq \int da \, e^{-e^{S_0} \operatorname{STr}\left(\frac{a^3}{3} + \hat{\zeta} a\right)}, \tag{4.9}$$

which is a variant of the Kontsevich model [46]. Equivalent derivations of this model from double-scaled matrix theories have appeared before in [39, 47], although not in terms of a supersymmetric formalism as we have utilized here. In the absence of sources, $\hat{\zeta} = \zeta \mathbb{1}$, the action is invariant under graded general linear transformations $a \to T a T^{-1}$, $T \in \mathrm{GL}(2|2)$, and the Efetov-Wegner theorem secures the normalization of the partition sum $\mathcal{Z}(\zeta) = 1$.

In the presence of sources, a parameterization of $a = \left( \begin{smallmatrix} \lambda_1 & \rho \\ \sigma & i\lambda_2 \end{smallmatrix} \right)$ followed by integration over the Grassmann variables $\rho, \sigma$ leads to

$$\mathcal{Z}(\hat{\zeta}) = \left( \frac{\partial}{\partial \zeta_1} - \frac{\partial}{\partial \zeta_2} \right) \int \frac{d\lambda_1 d\lambda_2}{2\pi} e^{-e^{S_0} \text{STr}\left( \frac{\lambda^3}{3} + \hat{\zeta}\lambda \right)}, \tag{4.10}$$

where the diagonal matrix $\lambda \equiv a|_{\rho=\sigma=0}$ contains the commuting variables, $\lambda_1, \lambda_2$ and a choice of integration contours safeguarding convergence is implicit to the definition of the integral. Finally the factor $(2\pi)^{-1}$ ensures the proper normalization of the bosonic measure, which we did not explicitly specify in Equation (4.4) above. In order to obtain the spectral density Eq. (2.9) with $z \to \zeta$, a further $\zeta_2$ derivative has to be taken, before setting the energy arguments equal to each other. This results in the Airy density of sates

$$\begin{aligned} \rho(\zeta) &= e^{2S_0/3}\left( -\text{Ai}(x)^2 + x\text{Ai}'(x)^2 \right), \qquad \left( x = -e^{2S_0/3}\zeta \right) \\ &\sim \frac{e^{S_0}}{\pi}\sqrt{\zeta}, \end{aligned} \tag{4.11}$$

where in the second line we have used the standard asymptotics of the Airy function (see Appendix C) to derive the leading behavior for $e^{2S_0/3}\zeta \gg 1$. It is then clear that the saddle-point evaluation we performed above is precisely the semi-classical limit $e^{S_0} \gg 1$ of the exact expression corresponding to the well-known asymptotics of the Airy function.

In principle, we may upgrade the above integral to one over GL(2|2) matrices to obtain the correlations in the Airy DoS right at the edge. However, for the present purposes, it is sufficient to move somewhat into the double scaled spectrum and expand around the symmetry broken stationary points Eq. (4.7) at finite $e^{-S_0} \ll \zeta \ll g$. In this case, there is nothing left to be done, and we can cut and paste from section (3), up to and including the final result, the sine-kernel level correction Eq. (2.26). The fact that the present model is scaled to the vicinity of the edge only makes its appearance in the scaling variable, $s = \pi\omega/\Delta$, which now makes reference to the near edge level spacing, $\Delta = \rho^{-1} = \pi e^{-S_0}\zeta^{-1/2}$.

Before concluding this section, let us comment on a point that will become relevant in connection with a minimal string theory interpretation in the next section. We derived the representation Eq. (4.9) by fluctuation expansion of the action around $\zeta = 0$, right at the edge where the mean field symmetry breaking comes to an end by definition. On the other hand, we know that for any $\zeta > 0$ mean field causal symmetry breaking is equivalent to the statement of a finite density of states on average. We can invoke this mechanism by a 'second' stationary phase approximation, this time applied to the fluctuation action Eq. (4.9), i.e. by asking for the least action fluctuation configurations, $\bar{a}$, for given $\zeta > 0$. A straightforward variation leads to

$$\bar{a} = \pm i\sqrt{\zeta}, \tag{4.12}$$

and the differentiation in sources at the causal of these yields the mean field spectral density in agreement with Eq. (4.8). Comparing with the exact result, Eq. (4.11), we interpret the mean field result as the approximation of the Airy functions in the semiclassical limit $e^{S_0} \gg 1$.

### 4.1.2 Disks, annuli and instantons

Having described the physics of spectral branes in terms of a double-scaled matrix model, we would now like to make the connection to the continuum approach of minimal string theory, which serves as the bulk description in this setting. We start by discussing a single determinant operator insertion. As described in [38], in the bulk the density of states is related to a disk amplitude with boundary condition given by a single energy variable $\zeta$, that is a worldsheet

with a single FZZT brane boundary condition $\zeta$, referred to in this context as the boundary cosmological constant. The connection between determinant operator insertion and spectral density is made by a reduced version of the flavor matrix integral (4.9)

$$\Psi(\zeta) := \langle \det(\zeta - H) \rangle = \int da \, e^{-e^{S_0}\left(\frac{a^3}{3} + \zeta a\right)} = e^{-S_0/3} 2\pi i \mathrm{Ai}\left(-e^{2S_0/3}\zeta\right), \qquad (4.13)$$

where $a, \zeta$ are single scalar variables (the former representing a one-dimensional flavor matrix) and in the last equality we noted that for an appropriate choice of integration contour the integral takes the form of a well-known integral representation of the Airy function (see Appendix C for details.). We interpret this representation as the effective theory of an FZZT brane upon integrating out the contribution of the ZZ branes, in terms of a single degree of freedom $a$.

At the end of the previous section we have seen that in the semiclassical evaluation of this miniature flavor theory we encounter two saddles, Eq. (4.12), a toy model version of the standard saddle and the Altshuler-Andreev saddle. Summation over these saddles and the fluctuations around them produced a large $\zeta$ semiclassical approximation of the spectral determinant, and the full $a$-integral an restoration of the exact result Eq. (4.13). Translating to the bulk, the double saddle point structure means that the brane exhibits two branches, where the second appears on the brane as an instanton. In the literature [39], this saddle point structure has been interpreted as a semiclassical approximation to the exact quantum target space of the brane, with $\zeta$ playing the role of an effective semiclassical target space coordinate.

So far we only considered a single determinant, which is not rich enough to exhibit Goldstone modes associated to symmetry breaking. In order to get from the toy model to the full problem, we need to consider more than one FZZT brane. From the perspective of this paper, the most natural realization would be one corresponding to the ratio of spectral determinants, with an effective supersymmetric flavor matrix representation. However, absent a matching bulk picture[12] we engage the replica formalism whose starting point are $R$-fold replicated determinants $\left\langle \det(\zeta_1 - H)^R \det(\zeta_2 - H)^R \right\rangle$. The matrix theory implementation of these products in terms of $R$-fold replicated Grassmann integrals is straightforward. The construction of section 3 modified for $2R$-Grassmann integration variables and zero commuting ones leads to a flavor matrix model (4.6) where $a$ is now a $2R \times 2R$ matrix. In the double scaled limit, this leads to (4.9) from where the non-perturbative structure of spectral correlations is obtained via a replica version of the AA saddle point detailed in Appendix B.

An alternative approach, more closely geared towards a bulk interpretation, starts from the representation of the double scaled expectation value, [39, 48]

$$\left\langle \prod_{i=1}^{R} \det(\zeta_i - H) \right\rangle = \frac{\Delta(d)}{\Delta(\zeta)} \prod_{i=1}^{R} \Psi(\zeta_i), \qquad (4.14)$$

where the 'wavefunction' $\Psi(\zeta)$ is defined in Eq. (4.13) above. Here $\Delta(\zeta) = \prod_{i<j}(\zeta_i - \zeta_j)$ is the Vandermonde determinant, and similarly for $\Delta(d)$, where $d = e^{-S_0}(\partial_1, \ldots, \partial_R)$ contains derivatives acting $i^{\text{th}}$ coordinate $\zeta_i$ of the wavefunction $\Psi(\zeta_i)$. In continuum language, the individual contributions in (4.14) are associated to particular worldsheet contributions. In the semiclassical limit $e^{S_0} \gg 1$, the product $\prod_{i=1}^{R} \Psi(\zeta_i)$ becomes a product of WKB wavefunctions, each factor comprising a disk diagram as well as an annulus with both boundaries on the same

---

[12]This would require a more explicit microscopic understanding of the ghost branes of [40] in general, and specifically in terms of the continuum worldsheet approach of minimal string theory. In view of how much simpler and transparent the non-perturbative analysis becomes compared to the replica approach, this might be a rewarding investment.

brane [39], and the ratio $\frac{\Delta(d)}{\Delta(\zeta)}$ comes from exponentiated annuli with boundary conditions on different branes. Let us see explicitly how this works for the case of two determinants, that is two exponentiated loop operators (4.2),

$$\langle \Psi(\zeta_1)\Psi(\zeta_2)\rangle = \left\langle e^{\mathcal{W}(\zeta_1)} e^{\mathcal{W}(\zeta_2)}\right\rangle . \tag{4.15}$$

A single insertion of the loop operator is the matrix-model representation of a worldsheet with a single boundary, the so-called disk amplitude, $\text{Disk}(\zeta) = \langle\mathcal{W}(\zeta)\rangle$. A two-loop correlator gives the annulus diagram $\text{ann}(\zeta_1,\zeta_2) = \langle\mathcal{W}(\zeta_1)\mathcal{W}(\zeta_2)\rangle$. Then to leading semiclassical order we can write, [4,49],

$$\langle \Psi(\zeta_1)\Psi(\zeta_2)\rangle \simeq e^{\text{Disk}(\zeta_1) + \text{Disk}(\zeta_2) + \text{ann}(\zeta_1,\zeta_2) + \frac{1}{2}\left(\text{ann}(\zeta_1,\zeta_1) + \text{ann}(\zeta_2,\zeta_2)\right)}. \tag{4.16}$$

We note that the individual factors contributing to this answer are precisely the ingredients we introduced above, namely disk diagrams with a boundary on either of the spectral branes, $\text{Disk}(\zeta_{1,2})$ annulus diagrams with both boundaries on the same brane, $\text{ann}(\zeta_{1,2},\zeta_{1,2})$, and finally an annulus diagram with one boundary on each spectral brane, $\text{ann}(\zeta_1,\zeta_2)$.

In the next section, we will analyse the connected annular contribution $\text{ann}(\zeta_1,\zeta_2)$ and extract the singular (in energy) terms coming from the EFT predictions in Section 2.5 above.

## 4.2 Universal RMT contributions in 2D gravity

In this section, we show how universal contributions of the EFT of chaos in Section 2.6 are recovered from the bulk perspective. To this end we evaluate the worldsheet diagrams contributing to (4.14) in the replica limit $R \to 0$ and match them to individual contributions in Section 2.5. Carrying out the disk and annulus expansion, (4.16) for the replicated determinants, one finds

$$W(\zeta_1,\zeta_2) = \partial^2_{\zeta_1,\zeta_2} \lim_{R\to 0} \frac{\left(\Psi(\zeta_1)^R - 1\right)\left(\Psi(\zeta_2)^R - 1\right)}{R^2} = \partial^2_{\zeta_1,\zeta_2}\text{ann}(\zeta_1,\zeta_2), \tag{4.17}$$

i.e. the only surviving connected contribution to the correlator of two resolvents, (2.4), is indeed the annulus $\text{ann}(\zeta_1,\zeta_2)$. This result is obvious once we recall that one can find the two-resolvent expectation value simply by computing the expectation value of two loop operators, $\langle\mathcal{W}(\zeta_1)\mathcal{W}(\zeta_2)\rangle$, which is nothing but our earlier definition of $\text{ann}(\zeta_1,\zeta_2)$ (see Figure 3). However it is reassuring to see that it also follows from the full machinery developed above. In particular, it would be impossible to develop our causal symmetry breaking analysis without full recourse to the determinants (4.14). We have thus established that we can extract the leading RMT singularity $\sim s^{-2}$ in bulk language from an annulus diagram with two FZZT boundaries, $Z(\zeta_1,\zeta_2)$, the so-called FZZT annulus partition function in minimal string theory. This amounts to computing the contribution of the Goldstone modes to the leading singularity around the standard saddle in the bulk, i.e. the exact bulk equivalent of the EFT diagram (2.22).

We thus want to compute the bulk annulus diagram, which in the worldsheet setting amounts to adding contributions from worldsheet ghosts $Z_{\text{ghost}}$, the 2d Liouville partition function $Z_{\text{Liouville}}$ as well as the $(2,p)$ minimal model CFT, $Z_{\text{matter}}$, leading to the final result,

$$Z_{(2,p)}(\zeta_1,\zeta_2) = \int_0^\infty dP \frac{\cos(4\pi s_1 P)\cos(4\pi s_2 P)}{2\pi P \sinh 2\pi\frac{P}{b}\cosh 2\pi\frac{P}{b}}, \tag{4.18}$$

which is the $(2,p)$ version of what we called $\text{ann}(\zeta_1,\zeta_2)$ above, and where

$$\zeta_{1,2} = \sqrt{2\pi}\kappa\cosh 2\pi b s_{1,2}, \qquad b = \sqrt{\frac{2}{p}}, \qquad \kappa = \sqrt{\frac{\mu}{\sin^2\pi b^2}}. \tag{4.19}$$

$$\left\langle \mathrm{Tr} \frac{1}{\zeta_1^+ - H} \mathrm{Tr} \frac{1}{\zeta_2^- - H} \right\rangle_{(2,p)}^{\mathrm{annulus}} = \partial^2_{\zeta_1, \zeta_2} \qquad \sim s^{-2}$$

Figure 3: The annulus contribution to the spectral two-point function. Its universal part gives the correct EFT singular contribution of $-1/2s^2$. The diagram is identical to (2.22), although here it represents a minimal-string worldsheet interpreted as a 2D spacetime with two branes, also known as the Laplace transform of a Euclidean wormhole. The 2D gravity amplitude gives the same leading singularity including the numerical coefficient as (2.22), coming from the Goldstone sector of causal symmetry.

The constant $\mu$ is the bulk cosmological constant of 2D gravity and we have chosen the normalization of the energy variables $\zeta_{1,2}$ so as to reproduce the leading density of states (4.8). A detailed exposition of the relevant minimal-string calculations can be consulted in [44, 45, 50].

### 4.2.1 The ramp

The first non-trivial contribution to the spectral correlation function is obtained by differentiating the FZZT annulus partition function, (4.18), twice with respect to the energy arguments

$$W^{\mathrm{annulus}}_{(2,p)}(\zeta_1, \zeta_2) = \partial^2_{\zeta_1, \zeta_2} Z_{(2,p)}(\zeta_1, \zeta_2), \tag{4.20}$$

which as we explained above is just the two-resolvent correlation function, now written in 2D gravity language (cf. Fig. 3). The differentiation with respect to energy arguments proceeds implicitly, using (4.19) and the resulting integral can be carried out explicitly, with the answer, [38, 44, 50],

$$W^{\mathrm{annulus}}_{(2,p)}(\zeta_1, \zeta_2) = \frac{\mathrm{sech}(b\pi s_1)\mathrm{sech}(b\pi s_2)}{32\pi\kappa^2 \left(\cosh(\pi b s_1) + \cosh(\pi b s_2)\right)^2}. \tag{4.21}$$

This finally gives the simple expression,

$$W^{\mathrm{annulus}}_{(2,p)}(\zeta_1, \zeta_2) = \frac{1}{4} \frac{1}{\sqrt{-\zeta_1 + \kappa}\sqrt{-\zeta_2 + \kappa}} \frac{1}{\left(\sqrt{-\zeta_1 + \kappa} + \sqrt{-\zeta_2 + \kappa}\right)^2}. \tag{4.22}$$

The spectral correlation function is then obtained via

$$\begin{aligned} R_2(\zeta_1 - \zeta_2) &= \frac{\Delta^2}{2\pi^2}\mathrm{Re}\, W(\zeta_1^+, \zeta_2^-)_{\mathrm{annulus}} + \mathrm{reg.} \\ &= -\frac{\Delta^2}{2\pi^2(\zeta_1 - \zeta_2)^2} = -\frac{1}{2s^2}. \end{aligned} \tag{4.23}$$

The terms we left out are regular in the limit $\zeta_1 \to \zeta_2$, as indicated in the first line. Taking the real part is equivalent to fixing a prescription to analytically continue the function $W(\zeta_1, \zeta_2)_{\mathrm{annulus}}$, which leads to the singular behavior we indicated. This is the bulk counterpart of the cylinder contribution in the topological expansion of the EFT of chaos given in Eq.

(2.22) and it is reassuring that the singular part of the diagram indeed coincides with that prediction. In this way it becomes clear how to map the advanced (red) and retarded (blue) boundaries in that representation to the properties of the FZZT boundary state given by the boundary cosmological constants $\zeta_{1,2}^{\pm}$ (see Figure 3).

While higher-genus contributions, such as (2.24) in symmetry classes different from the unitary one might be possible to compute in principle (but difficult in practice), the bulk interpretation of the Althshuler-Andreev saddle remains elusive, although it seems promising to attempt to construct these contributions from applying the Riemann-Siegel lookalike construction to contributions like (4.22). This would amount to giving a bulk semiclassical description of the leading singularity around the Altshuler-Andreev saddle, which we discussed from the EFT perspective in (2.25). The same is true for the physics of spectral correlations in the far infrared, where the level rigidity symptomatic for systems with hard quantum chaos reflects in a restoration of causal symmetry and integration over the full flavor matrix manifold. Since the non-perturbative structure of the EFT is most accessible in the supersymmetric formulation, it would be desirable to study the extension of the string worldsheet formalism to one that is dual to the supersymmetric (i.e. graded) flavor framework.

## 4.3 A three-dimensional example: $\mathbb{T}^2 \times I$ wormholes

In order to further illustrate the universality of the EFT, let us briefly highlight how the leading $1/s^2$ singularity appears in three dimensional gravity. The relevant computations are described in references by Cotler and Jensen, [13, 51], who give a path-integral quantization of three dimensional gravity on spacetimes with the topology of a torus times an interval, $\mathbb{T}^2 \times I$. This is the 3D analogue of the annular geometry in the minimal string theory above and we therefore expect to be able to extract the information of the EFT diagram (2.22) from this. We start with the expression of the two-boundary partition function, [13],

$$Z_{\mathbb{T}^2 \times I}(\tau_1, \tau_2) = \frac{1}{2\pi^2} Z_0(\tau_1) Z_0(\tau_2) \sum_{\gamma \in \mathrm{PSL}(2;\mathbb{Z})} \frac{\mathrm{Im}(\tau_1)\mathrm{Im}(\tau_2)}{|\tau_1 + \gamma\tau_2|^2}, \tag{4.24}$$

where $\tau_1$ and $\tau_2$ are modular parameters of the two tori, $\gamma$ is a Möbius transformation on $\tau_2$ and $Z_0(\tau) = \frac{1}{\sqrt{\mathrm{Im}\,\tau}|\eta(\tau)|^2}$ in terms of the Dedekind eta function. One can show that the leading low-temperature behavior at fixed spins $s_{1,2}$ on each of the boundaries gives, [13]

$$Z_{\mathbb{T}^2 \times I}(\beta_1, \beta_2) \sim \frac{1}{2\pi} \frac{\sqrt{\beta_1\beta_2}}{\beta_1 + \beta_2} e^{-E_1\beta_1 - E_2\beta_2} + \cdots. \tag{4.25}$$

This expression can be analytically continued to real time by setting $\beta_{1,2} \to \beta \pm iT$, which was used in [13] to obtain the linear ramp behavior. Since we are interested in the partition function at fixed energies $E_{1,2}$ instead, we first Laplace transform this expression with respect to the two temperature parameters, resulting in

$$Z_{\mathbb{T}^2 \times I}(E_1, E_2) = \frac{1}{2\pi} \int_0^\infty \frac{\sqrt{\beta_1\beta_2}}{\beta_1 + \beta_2} e^{-E_1\beta_1 - E_2\beta_2} = \frac{1}{4\sqrt{E_1}\sqrt{E_2}(\sqrt{E_1} + \sqrt{E_2})^2}. \tag{4.26}$$

Note the structural similarity with the expression (4.22) above. This allows us to immediately deduce the contribution of this diagram to the spectral two point function

$$\begin{aligned} R_2(\omega) &= \frac{\Delta^2}{2\pi^2} \mathrm{Re}\, Z_{\mathbb{T}^2 \times I}(E_1^+, E_2^-) + \mathrm{reg.} \\ &= -\frac{\Delta^2}{2\pi^2(E_1 - E_2)^2} = -\frac{1}{2s^2}, \end{aligned} \tag{4.27}$$

which gives the universal ramp contribution to the spectral form factor predicted by the Goldstone diagram (2.22). Again, the question of how to extend the bulk result to the non-perturbative level (i.e. $\mathcal{O}(e^{-L})$) remains open.

## 5 Discussion

The main theme in this paper has been the universal nature of chaotic spectral correlations governed by a symmetry-based effective field theory. One of the attractive features of this approach is that it applies both to ensembles, such as random matrix theory with an arbitrary invariant potential, or disorder-type models, such as the SYK model, but crucially also to individual theories without reference to ensembles or disorder averages. In the latter case the content of the effective field theory is to be understood as applying to the individual quantum system by providing an envelope function of the typical behavior of connected correlation functions or similar observables, where the true correlation function fluctuates potentially quite wildly around the envelope. The exact envelope behavior can be recovered by some 'mild' averaging, say a running time average of a real-time correlation function or an average over judiciously chosen energy intervals for a momentum-space correlation function. At the same time we have shown that specific gravity configurations (e.g. two-sided Euclidean wormholes) allow us to compute the universal enveloping function from the gravity perspective. One may thus entertain the hope that similar bulk solutions can likewise be associated to the computation of connected spectral correlations in higher-dimensional AdS/CFT pairs, such as the ABJM theory [52] or $\mathcal{N} = 4$ SYM theory [53] without having to construct ensembles of boundary theories (other than perhaps via some 'mild' averaging over a small set of parameters, e.g. moduli or coupling constants). In this way the EFT perspective gives a quite general justification to consider bulk wormhole-type contributions even for individual boundary theories, as would be desirable from the perspective of the recent bulk computations of the unitary Page curve from 'replica wormholes' [27,54–56] (see also [57] for a review emphasizing the many-body aspects as relevant to this work).

The idea that wormhole-type correlations are to be associated with the universal behavior of quantum chaotic systems has been investigated from the perspective of the eigenstate thermalization hypothesis in [58] and from the point of view of a conjecture of the statistics of the operator-product coefficients in [59]. It would be interesting to investigate further the relation between these approaches and ours. For this it may be useful to note that FZZT boundary conditions of the type discussed in Section 4.1 also play a role in establishing eigenstate thermalization in simple holographic models [60]. While it is typically hard to establish that a given theory or class of theories satisfies the eigenstate thermalization (the recent results of [60–62], as well as [63] from the gravity perspective notwithstanding), we have established here results of a morally very similar nature, which are however of a wide – in fact in a technical sense universal – applicability.

In cases where the effective field theory came from an underlying individual quantum system there is a sense in which correlation functions of the 'multi-boundary' type should refactorize into individual 'single-boundary' contributions, and it would be interesting to investigate how this can be incorporated into the EFT of chaotic quantum systems. Progress in this direction in JT gravity as well as minimal string theory, albeit without recourse to causal symmetry, was presented recently in [49,64].

It would be an important goal to examine higher-dimensional examples, and in particular to compare with recent attempts to associate ensemble-type interpretations to the bulk path integral. Our symmetry-based approach gives a clear picture in what sense it is appropriate to interpret fixed individual quantum systems with ensembles. Using the classification result

of Refs. [65, 66] together with the EFT Lagrangian (2.15) adapted to the correct symmetry class gives a direct algorithm to identify and write down the appropriate ensemble for any boundary theory of interest. Since the higher-dimensional bulk story in full generality seems to require a third quantized "Universe Field Theory", (see discussions in [27, 67–69]), a more modest intermediate goal might be to calculate the coefficients of universal singular diagrams predicted by our EFT from individual bulk manifolds, as we have done here for two and three-dimensional gravity.

## Acknowledgements

We would like to thank Alexandre Belin, Jan de Boer, Jie Gu, Thomas Guhr, Daniel Jafferis, Marcos Mariño, Pranjal Nayak, Francesco Riva, Steven Shenker, Douglas Stanford, Manuel Vielma for enlightening discussions regarding the contents of this paper, and especially Pranjal Nayak and Manuel Vielma for ongoing collaborations on related matters. This work has been supported in part by the Fonds National Suisse de la Recherche Scientifique (Schweizerischer Nationalfonds zur Förderung der wissenschaftlichen Forschung) through Project Grants 2000 20_ 182513, the NCCR 51NF40-141869 The Mathematics of Physics (SwissMAP), and by the DFG Collaborative Research Center (CRC) 183 Project No. 277101999 - project A03.

## A Expansion in Goldstone modes vs. topological recursion

As mentioned in the main text, the sigma model approach is applicable to a wide class of chaotic Hamiltonians, including to matrix Hamiltonians $H = \{H_{\mu\nu}\}$ drawn from distributions $P(H)dH$. Referring to section 3 for a technically detailed discussion, we here stay on a qualitative level and compare the EFT approach to the topological (genus) expansions central to traditional matrix field theory. It is instructive to formulate this comparison in a diagrammatic language, where we will us two slightly different languages for the representation of the theory's microscopic building blocks depicted in Fig. 4. On the left, we have the double line notation standard in matrix field theory, where individual matrix elements $H_{\mu\nu}$ are represented by parallel stubs, and their Wick contraction is indicated by a double line. The diagrammatic codes standard in the theory of disordered systems — which are equivalent but arguably are more efficient in conveying the information relevant to us — the same contraction is indicated by a single dashed line, as indicated on the right.

Presently, these building blocks appear in the context of the advanced and retarded resolvents $G(z^+) = (z^+ - H)^{-1}$ and $G(z'^-) = (z'^- - H)^{-1}$, which we may consider formally expanded in $H$. Upon averaging, this then leads to structures as indicated in Fig. 5 in double line (left) or impurity (right) syntax. Each diagram contributes to the perturbative expansion with a factor

Figure 4: Alternative ways of representing building blocks of matrix theories. Left: double line notation customary in matrix field theory. Right: 'impurity diagram representation' customary in the physics of random systems.

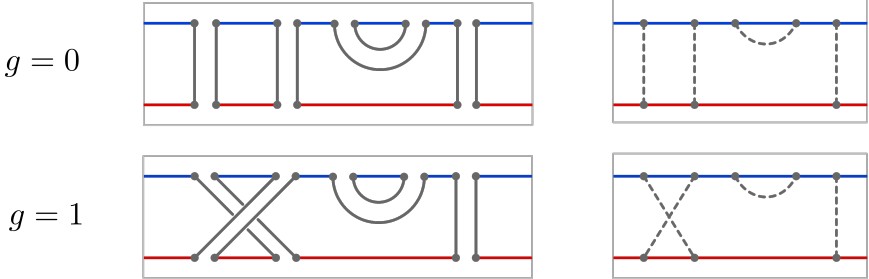

Figure 5: Alternative ways of representing building blocks of matrix theories. Left: double line notation customary in matrix field theory. Right: 'impurity diagram representation' customary in the physics of random systems.

$\sim L^{-g}$, where in matrix theory parlance the power $g$ measures its degree of non-planarity, or genus (cf. the upper and lower left panel), and in that of condensed matter theory the number of crossing impurity lines (upper and lower right panel).

For certain observables, topological recursion formulae establish quantitative connections between the total contribution of diagrams of genus $g + 1$ to that of one degree lower, $g$. This machinery has been engaged in Ref. [4] to establish a boundary–bulk correspondence by identifying equivalent hierarchies in the topological expansion of JT gravity purportedly equivalent to a matrix integral at the boundary. The perturbative approach to chaos or disorder in condensed matter physics is somewhat different: Its starting point is the observation that a tiny (exponential in $L$) subset of all diagrams contributing to a specific genus class is capable of producing a singularity in $\omega^{-g}$, where $\omega = |z - z'|$. We emphasize that the degree of the most violent singularity is precisely matched by the genus, so that the effective perturbative parameter of a series retaining only the maximally singular diagrams reads $1/(N\omega)^{g+l} \sim 1/s^{g+l}$, where $l$ is fixed and depends on the specific definition of the observables. (For example, $l = 2$ in the case of the spectral two point correlation function.) This is the expansion parameter featuring in the expansion of the EFT, indicating that the singular diagrams afford an interpretation in terms of Goldstone mode propagators and vertices of the field theory. To see this in more explicit ways, consider the three examples of singular diagrams contributing to the spectral correlation function (the closed fermion loops representing the trace over resolvents) shown in the left column of Fig. 6

The distinguishing feature of these diagrams is that they (a) define a subset of much higher order (lower entropy) compared to the full contents of a given genus class, (b) can be computed in closed form for matrix models or other 'microscopically' defined models of sufficient simplicity, and universally yield equal contributions to observables. For example, the figure $\infty$ diagram featuring in the second row appears in random matrix theory, the periodic orbit approach to quantum billiards (the Sieber-Richter contribution [70]), and the theory of disordered metals (weak localization correction, see Ref. [71] for a historic reference) as a contribution of identical topology but differently defined microscopic details. The actual computation of the diagrams from first principles can be tricky, equivalent in complexity to the construction of the EFT from a microscopically defined model (we will sidestep this complication for the time being.) However, in the deep infrared, they all contribute as $\sim s^{-(l+g)}$ with universal coefficients, and this topological equivalence of the most singular contributions across different theories is the key to understanding random matrix universality from semiclassical principles [24].

The EFT approach automatically filters out the singular diagrams and represents their contribution in terms of Goldstone mode fluctuations as discussed above. However, it is instructive

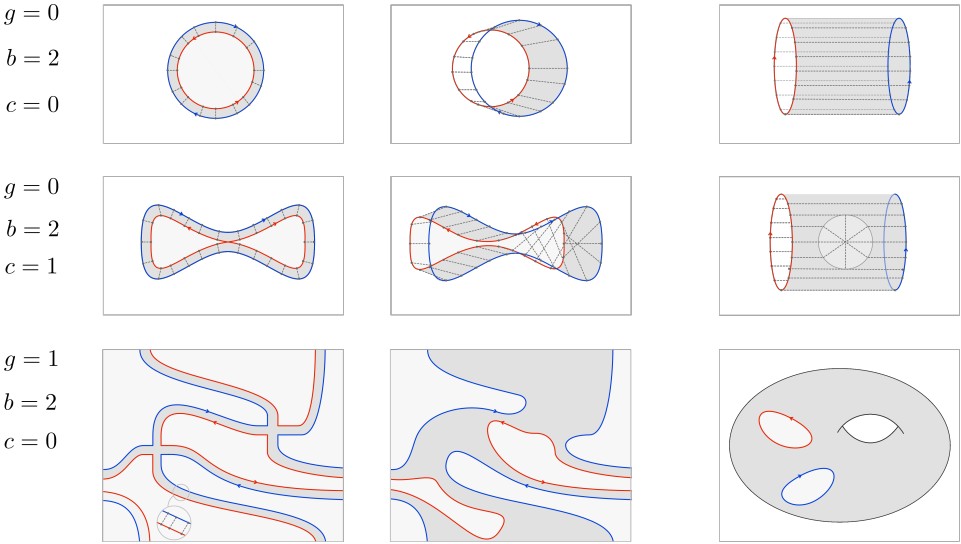

Figure 6: Impurity diagram representation of the perturbative surface to bulk correspondence. Each line is labelled by the genus $g$, the number of boundaries $b$ and the number of crosscaps $c$. Discussion, see text.

to stay for a moment on the microscopically resolved level of a matrix theory and interpret the correspondence indicated on topological grounds in Eqs. (2.22) to (2.24) in the microscopically resolved language. To this end, consider the three representatives of singular 'impurity diagrams' shown in the first column of the bottom part of the figure. Here, the first ('wagon wheel') has $g = 0$ and accordingly is singular in $s^{-2}$, translating to the linear in $\tau$ ramp upon Fourier transformation. (For later reference we note that the wheel comes in two incarnations, one with arrows in opposite orientation, and another, not shown and existing only in systems with time reversal invariance, with arrows in identical orientation.) Now pull the two rings forming the wheel apart to generate a cylinder covered with a pattern of parallel scattering lines. The latter defines a bulk surface describing the wheel shaped ribbon in different representation.

For a less trivial example, consider the $g = 1$ diagram which contains stretches of mutually parallel and anti-parallel propagator lines and hence exists only in systems with time reversal symmetry. Its higher loop order reflects in a stronger singularity, $\sim s^{-3}$, yielding a sub-leading correction $\sim \tau^2$ to the form factor of systems in the corresponding symmetry classes. This time, the rearrangement into two non-crossing propagator loops requires a section of maximally crossed statistical lines, which is information equivalent to the presence of a cross-cap in surface representation. Finally, the third diagram is one of two representatives with $s^{-4}$ singularity, mutually canceling each other in their contribution to the spectral form factor but not to other observables. While a consistently opposite arrow orientation indicates that time reversal symmetry is not essential, a representation avoiding the crossing of propagator lines requires an underlying torus topology (indicated by identified boundaries in the figure.) Again we may pull the propagator lines apart to end up with two loops connected by a surface that contains a handle and is covered with a fabric of non-intersecting scattering lines.

The high level of universality manifest in the topology of the diagrammatic expansion suggests the presence of a simple underlying principle. The latter becomes obvious once we return to the level of the EFT and abandon the perturbative approach in favor of a non-perturbative one.

# B  Supersymmetry vs replicas

Supersymmetry is the method of choice when it comes to the investigation of non-perturbative structures in quantum chaos. However, occasionally, supersymmetry is not an option and we must resort to the alternative formalism of replicas instead. Presently this happens in connection to our discussion of minimal string theory, where formulations dual to a matrix theory containing inverse determinants ('ghosts') are not available. We therefore must do with a fermionic, non-graded version of the theory.

While in the early days of the field replicas have been dismissed as ill suited to the extraction of any non-perturbative information [41], it later became clear that this verdict had been overly harsh. Beginning with Ref. [72], various non-perturbative effects in spectra and wave function localization have been successfully described by replica methods. To get the idea and touch base with the supersymmetry formalism, consider the product $\mathcal{Z} \equiv \prod_{i=1}^{R} \det(z_i - H)$ of $R$ determinants. Employing the first of $R$ energy arguments as a source, we obtain

$$\partial_{z_1} \mathcal{Z}\big|_{z_i=z} = \partial_{z_1}\big|_{z_1=z} \det(z_1 - H)\det(z - H)^{R-1} \xrightarrow{R \to 0} \frac{\partial_z \det(z-H)}{\det(z-H)}\,. \tag{B.1}$$

This expression teaches us that in the replica limit $R \to 0$ the fermionic partition sum assumes a form identical to that of the supersymmetric one Eq. (2.9), where the $R - 1 \to -1$ fermionic spectator determinants turn into a 'missing fermionic determinant' assuming a role of an bosonic one. Of course, all this presumes that the replica limit is well defined. Pushing on and subjecting the determinants to the field theory machinery we obtain an $R$-flavor theory, or $2R$ flavors if retarded and advanced resolvents are correlated with each other. At the saddle point level, we accordingly encounter $2^{2R}$ different choices of saddle point solutions. However, in view of the above fermion-boson analogy, one expects that only one of these assumes the role of the AA saddle. This is the one, where in each of the $2(R-1)$ spectators we pick the 'causal saddle', and in the sourced replica channel no. 1 invert the sign. Formally, one indeed finds that in the replica limit, the fluctuation contributions around all other saddles vanish. Adding the surviving replica AA saddle to the standard one, the correlation function (2.26) is recovered [72]. The recovery of the full result by saddle point analysis is again owed to the semiclassical exactness of the integrand. However, even for symmetry classes were this not fulfilling this criterion, an asymptotic expansion around the two saddle points yields excellent descriptions of spectral correlations to any desired precision.

# C  Airy function

In this appendix we review a few well-known facts regarding the Airy function and its complex integral representation which we made use of in the main text. Let us define the following integral

$$A_n(z) = \frac{1}{2\pi i} \int_{\mathcal{C}_n} e^{-\frac{t^3}{3}+zt} dt\,, \tag{C.1}$$

for any of the three contours shown in Figure 7. Each of these contours connects one allowed region (shown in grey in Figure 7) to another, where these regions are chosen such that the integrand decreases exponentially whenever one approaches infinity inside an allowed region. These are related to the standard Airy functions by

$$\mathrm{Ai}(z) = A_1\,, \tag{C.2}$$

$$\mathrm{Bi}(z) = i(A_2 - A_3)\,. \tag{C.3}$$

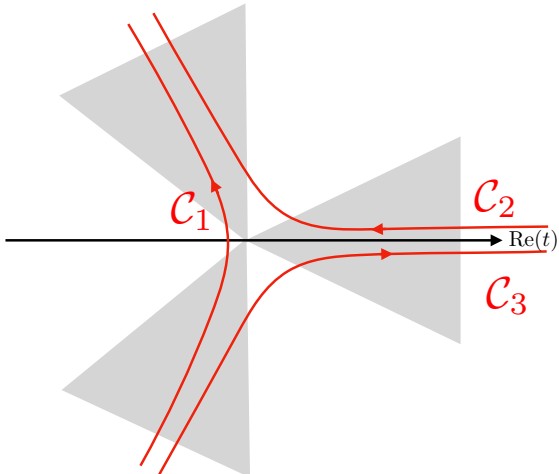

Figure 7: Choices of contours for the Airy integral. The allowed regions shown in grey with ranges $-\frac{\pi}{6} < \theta < \frac{\pi}{6}$, $\frac{\pi}{2} < \theta < \frac{5\pi}{6}$ and $-\frac{5\pi}{6} < \theta < -\frac{\pi}{2}$. Here we have defined $\theta = \arg t$. A 'good' contour approaches the point at infinity along one of the allowed directions and connects two good regions.

The choice contours and therefore which linear combinations or Airy functions are selected are dictated by the observable we wish to compute. We illustrate this on the example of the integral (4.10) in Section 4, that is

$$\mathcal{I} = \int \frac{d\lambda_1 d\lambda_2}{2\pi} e^{-e^{S_0} \operatorname{STr}\left(\frac{\lambda^3}{3} + \hat{\zeta}\lambda\right)}, \tag{C.4}$$

where we have the diagonal matrices $\lambda = \operatorname{diag}(\lambda_1, i\lambda_2)$, $\hat{\zeta} = \operatorname{diag}(\zeta_1, \zeta_2)$, and the integrations are initially over the real axis. The $\lambda_2$ integral is therefore deformable into the integral along the contour $\mathcal{C}_1$ and gives the standard Airy function Ai. The second integral, for $\lambda_1$ along the real axis must be deformed into either the contour $\mathcal{C}_2$ or the contour $\mathcal{C}_3$. Which contour is chosen depends on the sign of the imaginary part of $\zeta_1$. The correct contour choice is $\mathcal{C}_3$ for $\zeta_1 + i0$ and $\mathcal{C}_2$ for $\zeta_1 - i0$. All in all we then find that

$$\mathcal{I}_\pm = 2\pi \operatorname{Ai}(x_2)\left(\operatorname{Bi}(x_1) \pm i\operatorname{Ai}(x_1)\right), \qquad x_i := -e^{2S_0/3}\zeta_i, \tag{C.5}$$

where the subscript $\pm$ is correlated with the sign of the imaginary part of $\zeta_1$ in a hopefully obvious notation. Then the rest of the analysis in Section 4 goes through as advertised. We note that the fact that the contour of integration is chosen according to the imaginary part of $\zeta_i$ is a manifestation of the same mechanism which led to the choice of different saddles in the standard sigma model, but now at the level of the exact Airy integral, which is a special case of the (graded) Kontsevich model studied in Section 4.

We can furthermore use the integral representation (C.1) to give a simple description of the asymptotic behavior of the Airy function(s) for large positive and negative values of their argument. In order to obtain the asymptotic behavior of any of the three integrals defined in (C.1) one should perform the integral by steepest descent. For this we want to consider the solutions of the saddle-point equation $t^2 - z = 0$. For $z > 0$ we therefore have two saddle points on the real axis at $\pm\sqrt{z}$, while $z < 0$ leads to two saddles along the imaginary axis at $\pm i\sqrt{|z|}$. Which of the saddles contribute to a steepest descent analysis depends on the choice of original contour as well as the sign of the argument. Consider, for example the function

Ai($z$), that is the contour $\mathcal{C}_1$. For positive argument the steepest descent contour only passes through the saddle at $-\sqrt{z}$, while the saddle at $\sqrt{z}$ does not contribute. This leads to the well known asymptotics

$$\text{Ai}(z) \sim \frac{e^{-\frac{2}{3}z^{3/2}}}{z^{1/4}}, \qquad (z > 0). \tag{C.6}$$

However, if $z < 0$, the contour $\mathcal{C}_1$ is deformable into the steepest descent contour through both saddles at $t = \pm\sqrt{|z|}$, which leads to the oscillating behavior

$$\text{Ai}(z) \sim \frac{1}{z^{1/4}} \cos\left(-\frac{\pi}{4} + \frac{2}{3}|z|^{3/2}\right), \qquad (z < 0). \tag{C.7}$$

As we commented in the main text the same analysis goes through if we perform a semiclassical expansion of the Kontsevich model for $e^{S_0} \gg 0$, which we can understand by rescaling the Airy integrand (C.1)

$$e^{-e^{S_0}\left(\frac{\lambda^3}{3} - \zeta\lambda\right)} = e^{-\left(\frac{t^3}{3} - zt\right)}, \quad \Longleftrightarrow \quad e^{2S_0/3}\zeta = z, \, e^{S_0/3}\lambda = t, \tag{C.8}$$

so that the large $z$ asymptotic analysis just described exactly coincides with the $e^{S_0} \gg 1$ semiclassical evaluation of the integral, as advertised.

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
