# Peer review of "Late time physics of holographic quantum chaos"

_SciPost Physics, doi:SciPost Phys. 11, 034 (2021)_

## Round 2 · Referee Report · Anonymous (Referee 1) · 2021-1-12

Strengths

1- This paper very clearly reviews a formalism for understanding some universal behavior (statistics of energy levels) in chaotic quantum systems, with an emphasis on applications to quantum gravity. This formalism is not widely appreciated in the quantum gravity community, but is certainly of some interest to this community. Likewise, recent developments in this subject from the quantum gravity community may be of interest to the condensed matter/quantum chaos community, for whom the formalism described is more well known, but the applications to quantum gravity may not be. The paper does a good job of making the material accessible, and will likely be a valuable resource for researchers trying to understand progress made in the other field.

2- This paper also provides some novel applications of this formalism for 2D quantum gravity.

Weaknesses

1- Just a couple of very minor points/typos which I thought were unclear, which I list below.

Report

I think this is a very useful paper and worthy of being published in this journal.

Requested changes

1- The term "low energies" is sometimes used to mean small energy differences. While this is clarified on page 10, there are a couple of instances before this in which I think it might be unclear.

2- Throughout most of the paper the trace is denoted with lowercase tr, but on page 7 "Tr" is used.

3- In the figure near the top of page 16, I think some of the arrows might be facing the wrong direction.

4- A typo on page 17, second to last paragraph, it says "two two" instead of "to two".

5- On page 27, just below eq. 3.23, there are two epsilons when I think there should be one.

6- On page 31, in the last sentence of the caption for Figure 2, it says "s world-sheet" instead of "a world-sheet".

---

## Round 2 · Referee Report · Anonymous (Referee 2) · 2021-1-18

Strengths

1-The statement is clear. 2. It is multidisciplinary that makes it useful for a wide range of people of interest.

Weaknesses

1- There are few typos should be fixed.

Report

This paper is devoted to study certain universal nature of chaotic spectral correlations within
a framework of an effective field theory with symmetry breaking pattern.
An interesting question people are trying to explore is to see in what extend having the disorder average or ensemble averaging is crucial in this context.
In this paper the authors have argued
that their approach is applicable for both system with and without ensembles or disorder averages.
The main role is played by what is called causal symmetry breaking.

I found the paper very interesting and promising though a little bit hard to read for people who are not expert in the filed.
This might be the case mainly because of the nature of the subject, being multidisciplinary.
I think the paper has interesting impact on the field and therefore I recommend its publication.

---

## Round 3 · Referee Report · Anonymous (Referee 3) · 2021-7-2

Strengths

1- This paper very clearly reviews a formalism for understanding some universal behavior (statistics of energy levels) in chaotic quantum systems, with an emphasis on applications to quantum gravity. This formalism is not widely appreciated in the quantum gravity community, but is certainly of some interest to this community. Likewise, recent developments in this subject from the quantum gravity community may be of interest to the condensed matter/quantum chaos community, for whom the formalism described is more well known, but the applications to quantum gravity may not be. The paper does a good job of making the material accessible, and will likely be a valuable resource for researchers trying to understand progress made in the other field.

2- This paper also provides some novel applications of this formalism for 2D quantum gravity.

Weaknesses

Typos, etc, already addressed by authors.

Report

I think this is a very useful paper and worthy of being published in this journal.

The application of the approach to studying level statistics discussed on this paper (based on causal symmetry breaking) to holography, without recourse to ensemble-averages, is intriguing. This is a topic currently under active discussion, and this paper provides some valuable input.

---

## Round 3 · Author Response

Dear Editor, we would like to thank the referees for the thorough reading of our manuscript and for their helpful suggestions. Please find below a list of responses and changes addressing these comments. We hope these address all outstanding issues and allow the paper to be published in the present form. Regards, The Authors.

---

## Round 3 · List of Changes

- At the bottom of page 4 in the paragraph below Eq. (1.2), we added a further sentence clarifying the fact the low-energy in the EFT treated in this paper refers to small energy differences. In particular we added the sentence "We remark that, as will become clear below, the notion of `low-energy' in the EFT we develop here, refers to small {\rm differences} of energies in the spectrum of the original system."

- We have looked at the figure on top of page 16 and do not agree with the referee's comment regarding the direction of arrows, which are both in line with the convention in the literature (on flavour-space matrix models) and correspond to the contraction flow in the corresponding formulae.

- Addressing the issue of typos and notational consistency: we have gone once again over the entire manuscript and eliminated various lingering typos. In particular (but not exclusively) we have corrected all those pointed out by Referee 1. We thank them for their attentive reading. Please find below a list of all changes that are more significant than just adding or removing a plural 's' here and there. Other changes of the more 'trivial' kind are left uncommented.

- We updated the notation of various kinds of traces occurring throughout the manuscript to be fully consistent. We now use 'Tr' for ordinary traces and 'STr' for graded traces throughout. This also applies to all figures in which such traces appear.

- "two two" changed to "to two"

- one $\epsilon$ removed from text in paragraph just below (3.23)

- "s world-sheet -> a world sheet"

- one more important typo we noticed and fixed: in Eq. (2.24) the super trace structure of formula corresponding to the diagram shown was slightly wrong and has now been corrected.

---

## Editorial Decision

published